



# Achieving consistency between in-situ and remotely sensed optical and microphysical properties of Arctic cirrus: the impact of far-infrared radiances

Gianluca Di Natale[1], Helen Brindley[2], Laura Warwick[3], Sanjeevani Panditharatne[2,4], Ping Yang[5], Robert Oscar David[6], Tim Carlsen[6], Sorin Nicolae Vâjâiac[7], Alex Vlad[7,8], Sorin Ghemuleț[7], Richard Bantges[2], Andreas Foth[9], Martin Flügge[10], Reidar Lyngra[10], Hilke Oetjen[3], Dirk Schuettemeyer[3], Luca Palchetti[1], and Jonathan Murray[2]

[1]Consiglio Nazionale delle Ricerche, National Institute of Optics, Via Madonna del Piano, 10, Sesto Fiorentino, 19100, Firence, Italy
[2]Department of Physics and National Centre for Earth Observation, Imperial College, London, UK
[3]ESA/ESTEC, Noordwijk, the Netherlands
[4]RAL Space, Harwell Oxford, Chilton, UK
[5]Department of Atmospheric Sciences, Texas A&M University, CollegeStation, TX 77843, USA
[6]Department of Geosciences, University of Oslo, Oslo, Norway
[7]National Institute for Aerospace Research "Elie Carafoli", Bucharest, Romania
[8]University of Bucharest, Faculty of Physics, 405 Atomistilor, Magurele 077125, Romania
[9]Leipzig University, Leipzig Institute for Meteorology, Leipzig, Germany
[10]Andøya Space, Andøya, Norway

**Correspondence:** Gianluca Di Natale (gianluca.dinatale@ino.cnr.it)

**Abstract.** This paper describes the first retrieval of cirrus optical and microphysical properties from ground-based measurements simultaneously with co-located measurements from aircraft. In particular, the present effort exploits infrared radiances spanning the mid to far-infrared spectral regime based on co-located in-situ aircraft sampling and ancillary ground-based remote sensing. Spectrally resolved radiances covering the range 400-1500 cm$^{-1}$, in-situ measurements of cirrus particle sizes

and habits, backscatter ceilometer observations of cloud vertical structure and microwave inferred temperature and humidity profiles are used to investigate whether we can obtain consistency between the derived cloud properties and atmospheric state from these independent sources of data. The primary focus of this study is on the sensitivity of the retrieved cloud particle size to the assumed crystal habit. Excellent consistency of the retrieved cloud parameters is achieved both with the ceilometer derived optical depth and the size distribution measured by the aircraft by assuming the crystal habit to be comprised of

bullet rosettes. The averaged values of the effective diameter and optical depth obtained from radiometric measurements are (26.3±0.5) $\mu$m and (0.130±0.004) in comparison with the values derived from in-situ and ceilometer measurements equal to (31.5±5.0) $\mu$m and (0.120±0.004), respectively. Furthermore, we demonstrate that the radiance information contained within the far-infrared (wavenumbers < 650 cm$^{-1}$) spectrum is critical to achieving this level of agreement with the in-situ aircraft observations. The results emphasize why it is vital to expand the currently limited, database of measurements encompassing

the far-infrared spectrum, particularly in the presence of cirrus.



# 1 Introduction

Cirrus clouds play a vital role in regulating the energy balance of our planet and in determining how it, and hence Earth's climate, may evolve in the future (e.g. Lynch (1996), Dawson and Schiro (2025)). Most studies imply that cirrus imparts a net warming to the climate system (Choi and Ho, 2006) but that the degree of warming is dependent on the cloud fraction,

geographical location, altitude, and, crucially, optical thickness. Theoretical considerations and model projections point towards an increase in high cloud altitude in a warming climate (e.g. Hartmann and Larson (2002); Chepfer et al. (2002)), which, in the absence of any other change, would be expected to enhance the associated cloud radiative heating (Gasparini et al., 2024). Conversely, while observational studies of the response to inter-annual surface temperature variability indicate an increase in cirrus amount with warming (Zhou et al., 2014), model studies imply a reduction in the coverage of anvil cirrus with increasing

surface temperature via the 'iris-stability' hypothesis (e.g. Bony et al. (2016)). More recent observational studies imply a small tropical high cloud feedback due to compensating longwave and shortwave effects associated with changes in coverage, altitude and optical thickness (Raghuraman et al., 2024). Whatever response is ultimately realised, exactly how changes to cirrus coverage and location manifest radiatively will also critically depend on whether the associated cloud microphysics are also altered.

For example, previous work has shown the substantial impact of changes in particle size distribution on cirrus longwave heating rates, with the presence of smaller particles causing a sizeable increase in heating (e.g. Stackhouse and Stephens (1991)). Similarly, relatively simple parameterisations expressing cloud emittance as a function of effective radius have been used to demonstrate that the particle size of ice crystals determines if the presence of cirrus results in a positive or negative ice-water feedback and, consequently, an additional surface warming or the opposite depending on the sign of the feedback

(e.g. Stephens et al. (1990)).

A second difficulty in describing the interaction between radiation and cirrus is due to the myriad of different crystal habits which can exist in nature. Each shows different absorption and scattering behavior (Baran, 2009; Baran et al., 2014). Improved knowledge of the crystal habit distribution within cirrus, alongside measures of the crystal size and overall cloud optical depth is therefore critical to reducing uncertainty in their radiative effect.

Finally, to tie cirrus micro-physical and optical properties to their radiative impact we need to have confidence in the tools we use to simulate radiances and fluxes in the presence of these clouds. While a number of observational campaigns have taken place to assess cirrus 'radiative closure' over the visible, mid-infrared and microwave part of the Earth's electromagnetic spectrum (Meyer et al., 2016; Evans et al., 1999; Lawson et al., 2006; Turner and Mlawer, 2010), considerably fewer equivalent studies cover the far-infrared (FIR), typically defined as wavelengths between 15–100 $\mu$m . Simulations have long highlighted

the dominant contribution that we expect the FIR to make to the Earth's total outgoing longwave radiation and greenhouse effect (Brindley and Harries, 1998; Collins and Mlynczak, 2001), particularly for colder scenes. However, aircraft-based studies which attempted to reconcile FIR radiances measured in the presence of cirrus with simulations struggle to do so, even when using ice optical property databases that are routinely employed in satellite retrievals exploiting the mid-infrared and visible parts of the spectrum (Cox et al., 2007; Bantges et al., 2020), but that are still not validated in the FIR.



Part of the issue with these existing studies may be related to the experimental design. All employed a single aircraft platform, with in-situ sampling of the cloud microphysics necessarily occurring at a different time to the above-cloud radiance measurements. Hence, it is impossible to unambiguously map the cirrus properties to its radiative signature. An alternative to this approach is to use zenith-view radiances measured from the ground in the presence of cirrus. With this set-up, a very dry atmosphere is required to avoid the cirrus radiative signatures in the FIR being obscured by strong absorption by water vapour.

Exploiting such conditions, zenith radiances have been used to retrieve ice cloud properties over the Arctic and Antarctica from Extended and Polar Atmospheric Emitted Radiance Interferometers (E-AERI, P-AERI), extending down to 550 and 400 $cm^{-1}$, respectively, in a number of field campaigns (Maesh et al., 2001a, b; Turner et al., 2003; Garrett and Zhao, 2013; Shupe et al., 2013). Similarly, a multi-year, database of FIR radiances has been collected by the Radiation Explorer in the Far InfraRed - Prototype for Applications and Development (REFIR-PAD), operating autonomously at the Dome-C site Antarctica since 2011

(Palchetti et al., 2015; Bianchini et al., 2019). Several studies have exploited the data, developing methods to characterise cloud type and derive optical depth and size information (Di Natale et al., 2020). Most recently, observations from the Far Infrared Radiation Mobile Observation System (FIRMOS, Belotti et al. (2023)) taken from Zugspitze in the German Alps demonstrated the consistency of optical depth retrievals derived from zenith view infrared radiances with those estimated from simultaneous backscatter ceilometer observations (Di Natale et al., 2021). The development and deployment of FIRMOS was undertaken

specifically in preparation for the Far-infrared Outgoing Radiation Understanding and Monitoring (FORUM) Earth Explorer 9 mission (Palchetti et al., 2020).

Despite the notable advances made in these ground-based studies, including the key step of linking observations from complementary sensors to the measured infrared radiances, none include simultaneous in-cloud observations. In this paper we exploit a dataset collected as part of the MC2-ICEPACKS campaign, based at Andøya, Norway during February-March

2023 (David et al, in prep). The data include downwelling radiance observations made by the Far-INfrarEd Spectrometer for Surface Emissivity (FINESSE, Murray et al. (2024)) alongside independent measurements of the atmospheric vertical structure and, uniquely, contemporaneous in-situ sampling of cirrus microphysics from aircraft. We use these data to provide a well-characterised test of our ability to retrieve atmospheric temperature and water vapour profiles, ozone total columns and cloud optical and micro-physical properties given currently available parameterizations of ice crystals, with a particular emphasis on

demonstrating observationally how FIR radiances can constrain crystal habit.

Our manuscript is structured as follows: in section 2 we introduce the measurement field campaign and the instruments involved in the experiment; in section 3 we describe the methodology applied through the comparison of the retrieval products with the available measurements and the retrieval method applied; in section 4 we show and discuss the results and in section 5 we demonstrate the improvement in retrieval quality when FIR radiances are available to constrain habit selection. Finally,

in section 6 we draw conclusions and outline areas for future work.





## 2   MC2-ICEPACKS Campaign instrumentation

The Mixed-phase Clouds and Climate -Ice Pockets in Arctic Clouds at sub-Kilometer Scales campaign (MC2-ICEPACKS) took place from 13th February to 17th March 2023, based at Andøya, Norway, with a primary goal of investigating the distribution of Arctic cloud phase in order to improve its representation in Earth System Models (David et al, in prep). The campaign consisted of a ground-based segment and airborne measurements. The ground-based segment was hosted at both the Arctic LiDAR Observatory for Middle Atmosphere Research (ALOMAR) that is located on top of the Ramnan mountain 378 m a.s.l. and at Andøya Space (Fig. 1), located at 12 m a.s.l.. The airborne measurements were facilitated by the National Institute for Aerospace Research of Romania (INCAS) Atmospheric Research Laboratory aircraft that was stationed at Andøya Airport. The horizontal distance between ALOMAR and Andøya Space is approximately of 2.1 km.

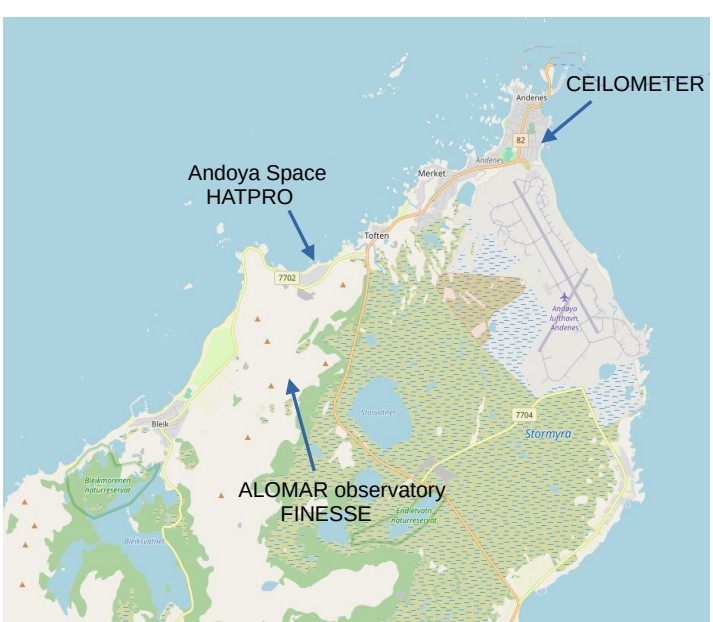

**Figure 1.** Location of ground-based instrumentation during the MC2-ICEPACKS campaign. FINESSE was located at an elevation of 378 m a.s.l., at ALOMAR [69.28°N, 16.01°E]. The HATPRO was installed at Andøya Space and the ceilometer close to Andenes (0 m a.s.l.). Background map: © OpenStreetMap contributors, data available under the Open Database License (ODbL). Retrieved from www.openstreetmap.org.

This paper concentrates on the measurements conducted on 17th February 2023, when thin, patchy cirrus clouds were observed over ALOMAR. These were sampled by the INCAS aircraft (section 2.1) between 08:15 and 08:45 UTC during an extended research flight (Fig. 2). Simultaneous ground-based observations were performed by FINESSE (section 2.2), a ceilometer (section 2.3, see Fig. 1) and a Humidity And Temperature PROfilers (HATPRO) radiometer (section 2.4) located at Andøya Space.





## 2.1 INCAS aircraft

The INCAS aircraft, a Hawker Beechcraft King Air C90GTx (INCAS KA), is equipped with two combination cloud probes mounted under the wings (e.g. Vâjâiac et al. (2021)), namely a HAWKEYE (SPEC, Boulder, CO) and a Cloud, Aerosol and Precipitation Spectrometer (CAPS; DMT, Boulder, CO)). The HAWKEYE includes a 2D-S, which combines two orthogonally oriented optical array probes with 10 and 50 $\mu$m resolution, respectively (spanning from 10 to 1280 and 50 to 6400 $\mu$m), a cloud particle imager (CPI) that takes 2-D images of cloud particles between 2.3-6400 $\mu$m and a Fast Cloud Droplet Probe for sampling the size distribution of smaller particles, from 1.5-50 $\mu$m (e.g., Lawson et al. (2001); Woods et al. (2018)). The CAPS provides similar aerosol and hydrometeor size information but due to an optical alignment issue during the presented case study, it was only used for observations of the true air speed, temperature and pressure during the flight. The ice crystal particle size distributions (PSDs) used here are calculated from the 50 $\mu$m -channel of the 2D-S using the System for OAP (optical array probe) Data Analysis (SODA2; developed at NCAR). During post-processing only ice crystals which were captured completely within the optical array ('all-in'; Heymsfield and Parrish (1978)) were considered and their size was calculated using the circle-fit method, or the diameter of the smallest circle that encompasses them (Wu and McFarquhar, 2016). The influence of shattered particles was removed following Field et al. (2006).

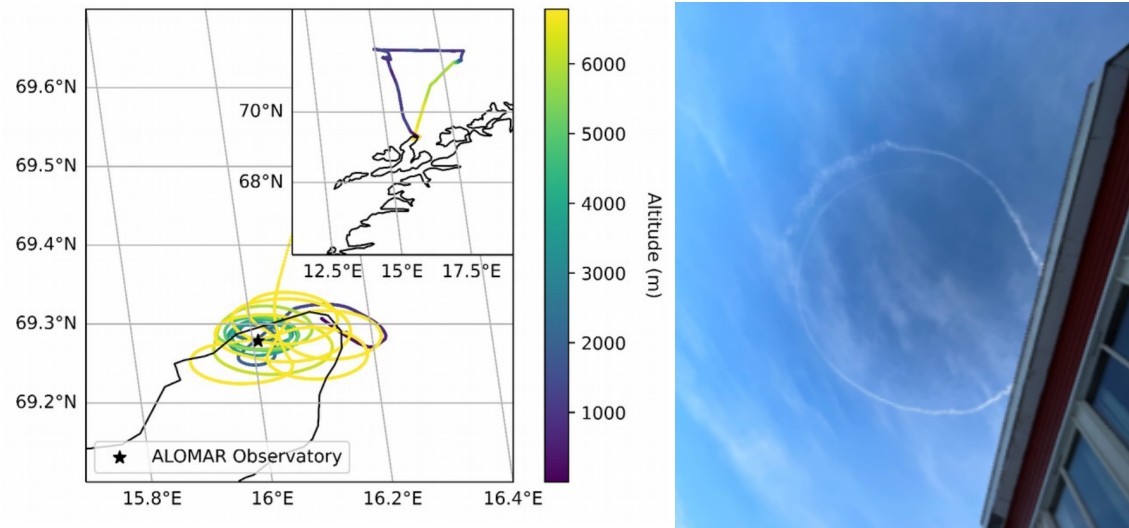

**Figure 2.** Left: Main image - Flight track of INCAS KA on 17th February 2023 over ALOMAR between 0800-0845 UTC. The colour shows the altitude of the INCAS KA. Inset – Trajectory of the entire science flight. Right: Sky conditions above FINESSE at 08:24 UTC during the INCAS KA overpass (credit J.Murray)

On 17th February the INCAS KA made several passes over ALOMAR with the aim of in-situ sampling of the cirrus cloud that was in the field of view (FOV) of FINESSE. The left panel of Fig. 2 shows the trajectory of the aircraft with the colour indicating the altitude of the aircraft. The right panel indicates its location above FINESSE just before the point of closest approach. Fig. 3 shows the variation in aircraft altitude and velocity during the flight with time. The airborne team used visual



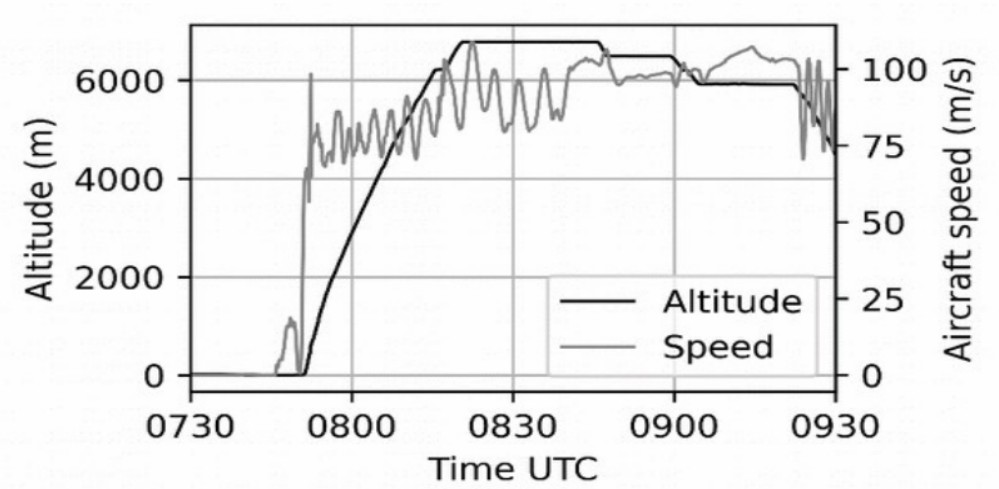

**Figure 3.** INCAS KA altitude and velocity as a function of time during the February 17th 2023 flight. The INCAS KA was over ALOMAR between approximately 08:15-08:45 UTC as the aircraft contrail was clearly visible from the ground.

and near-real time observations from the cloud probes to guide their flight level, implying that the cirrus above ALOMAR was at an altitude between 6-7 km with a peak at approximately 6.7 km.

## 2.2 FINESSE spectrometer

The FINESSE spectrometer (see Fig. 4 deployed outside the ALOMAR facility) is designed to measure spectrally resolved radiances from 400-1500 $cm^{-1}$ with a nominal spectral resolution of 0.5 $cm^{-1}$. The instrument consists of a commercial Bruker EM27 spectrometer coupled with a bespoke front-end pointing optics and calibration system. This enables the instrument to view in all directions from nadir through to zenith (Murray et al., 2024).

During the 17th February, the instrument was placed outside the ALOMAR building an hour prior to the aircraft overpass and operated in zenithal configuration, with measurements made between 07:30 and 12:00 UTC. The EM27 was configured to acquire interferograms in groups of 40 scans. One scan, at 0.5 $cm^{-1}$ resolution, takes 1.5 s to acquire. One fixed scene observation then takes 1 minute. The FINESSE control interface was initially set to acquire calibration scans of hot and ambient temperature blackbodies followed by 2 zenith views, this cycle of observations taking 4 minutes was repeated until just before the arrival of the aircraft at 08:15 UTC, when the number of zenith observations between calibrations was increased to 6.

Given the goal of the study the first step involved matching the FINESSE sky-views as closely as possible to the aircraft overflights. FINESSE has a beam diameter of 38 mm at its input window. This diverges with a full angle of 1.7°; therefore, the diameter of the FINESSE FOV at the cloud height was roughly 200 m. Given the INCAS aircraft speed of 85 m s$^{-1}$ (Fig. 3) this implies that in the ideal case the aircraft would traverse the FINESSE FOV in around 2.5 seconds. In reality, there




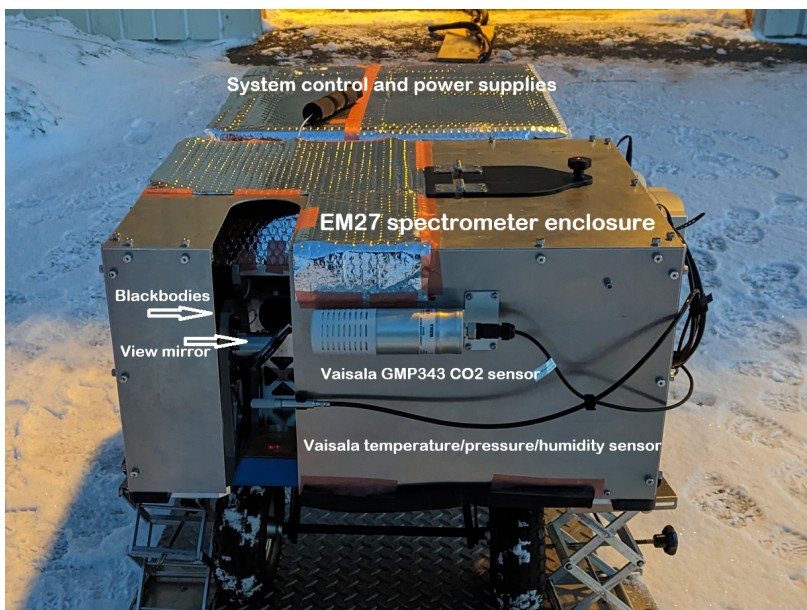

**Figure 4.** A front on view of FINESSE in its operational position outside the ALOMAR facility. The calibration blackbody targets, and scene selection mirror are seen within the spectrometer enclosure through the open system aperture. The EM27 input aperture is to the right of the mirror within the enclosure. Also highlighted are Vaisala sensors measuring the ambient atmospheric conditions. The system control and power supply unit is to the rear.

was always a slight mismatch between the location of the aircraft and the FINESSE FOV. For this analysis we consider all FINESSE spectra (obtained through the Fourier transform of the interferograms) that were taken when the aircraft location was within 1 km of the FINESSE FOV. Using this criterion, a subset of 11 spectra, temporally and spatially co-located with the other measurements, were selected between 8:25:23 UTC (30327 sec. since midnight, co-location 1) and 8:37:49 UTC (31069

sec. since midnight, co-location 11 (Table 1).

To help constrain the environmental conditions in the vicinity of FINESSE, two additional Vaisala probes are attached to the instrument casing. The Vaisala PTU303 probe measures pressure, temperature and relative humidity with quoted uncertainties of 0.1 mb, 0.3 °C and 1%, respectively. The Vaisala GMP343 probe records the $CO_2$ mixing ratio. We make use of the average value recorded during the FINESSE observing period of 395 ppmv to scale the profile used in the radiative transfer simulations

described in section 3.1. The temperature and humidity measurements are also used to help evaluate the FINESSE retrievals (section 4.2)

## 2.3    Ceilometer

The Met-Norway operated ceilometer (Lufft CHM15k) is permanently installed at the northerly tip of Andøya, approximately 6.3 km from ALOMAR and 4.4 km from Andøya Space (see Fig. 1). The ceilometer is designed to measure the backscattering



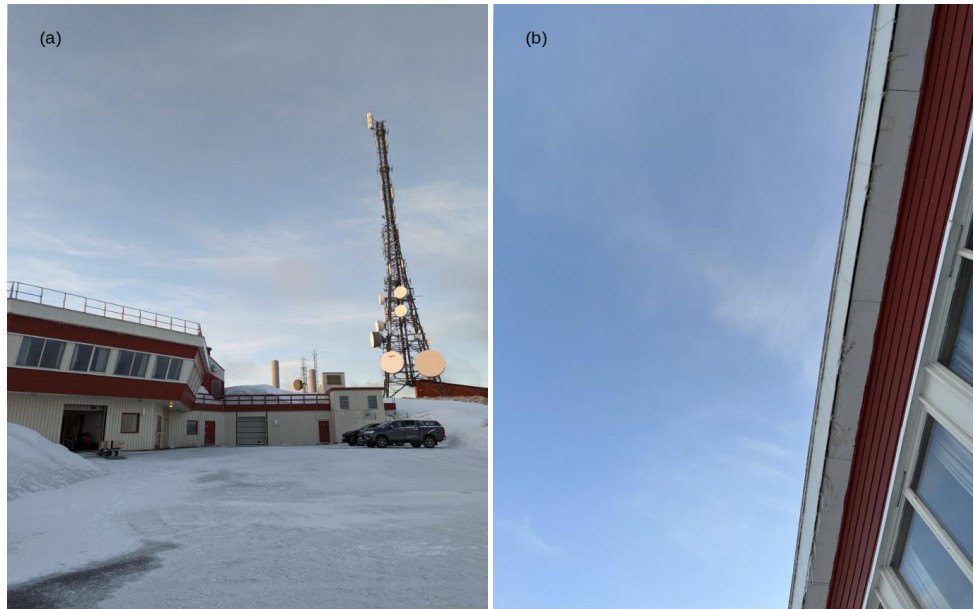

**Figure 5.** (a) View of ALOMAR with FINESSE visible right in front of the entrance of the facility (b) Passage of cirrus clouds over the site on 17th February 2023 at 07:45 UTC.

**Table 1.** Selected FINESSE spectra for the analysis

| Co-location index | time UTC (hh:mm:ss) | sec. since midnight |
|:---:|:---:|:---:|
| 1 | 08:25:23 | 30323 |
| 2 | 08:25:25 | 30325 |
| 3 | 08:25:27 | 30327 |
| 4 | 08:25:29 | 30329 |
| 5 | 08:25:31 | 30331 |
| 6 | 08:25:33 | 30333 |
| 7 | 08:37:41 | 31061 |
| 8 | 08:37:43 | 31063 |
| 9 | 08:37:45 | 31065 |
| 10 | 08:37:47 | 31067 |
| 11 | 08:37:49 | 31069 |

profile of aerosols and clouds. The passage of cirrus clouds over ALOMAR is pictured in Fig. 5. The left panel of Fig. 6 shows the detection of broken cirrus over the site on the 17th February 2023. During the time of the selected FINESSE observations (denoted by the black vertical dashed lines), above the boundary layer the strongest ceilometer returns emanate from between 6-7 km, peaking, in agreement with the aircraft observations, at around 6.7 km (Fig. 6, right panel). Even though the peak of





the signal is located a about 6.7 km, non-negligible signal is present at the lower layers, down to about 6 km of altitude. This
was also confirmed by the detection of ice crystals at 6 km by the probes on board the aircraft. These backscattering profiles
are used to fix the cirrus top and bottom heights (CTH and CBH) used in the radiative transfer simulations described in detail
in section 3.1.

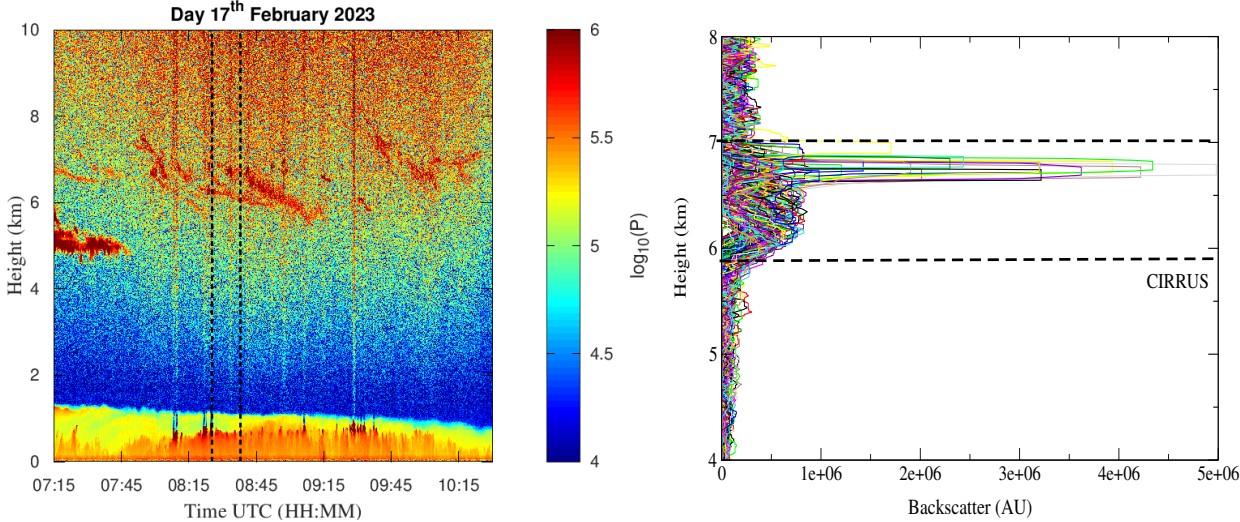

**Figure 6.** Left: Logarithmic backscattering signal ($P$, in arbitrary unit (AU)) detected by the Met-Norway ceilometer on 17th February 2023
showing the passage of cirrus clouds between 7:15 and 10:30 UTC. The dashed vertical black dashed lines at 8.25 and 8.38 UTC delimit the
time window of the FINESSE measurements selected for analysis. Right: Individual backscattering profiles obtained during the 8.25-8.38
UTC time window.

## 2.4   HATPRO radiometer

The Humidity And Temperature PROfiler version G4 (HATPRO; RPG Radiometric Physics GMBH, DE) is a microwave
radiometer designed as a network suitable low-cost passive instrument able to observe liquid water path (LWP), column water
vapour and humidity and temperature profiles with high temporal resolution of up to 1 s. The HATPRO is equipped with two
bands, K and V, one (K) in the range 22–31 GHz, and one (V) covering frequencies between 51-58 GHz (Löhnert et al., 2009).
Both the K and V bands are composed of seven channels. K provides humidity, column water vapour and LWP information
by exploiting pressure broadening of the 22.235 GHz $H_2O$ line. The V band is used to generate temperature information.
Comparisons of HATPRO retrievals made as part of the Multidisciplinary drifting Observatory for the Study of the Arctic
Climate expedition with radiosonde launches show agreement in integrated water vapour to within 0.35 kg m$^{-2}$ when values
are lower than 5 kg m$^{-2}$, conditions similar to those experienced during MC2-ICEPACKS. The same analysis suggests a
marked overestimate of humidity at the surface, switching to a small underestimate by 2-3 km. For temperature, biases were
of the order 0.7-1.8 K for the lowest 2 km of the atmosphere (Walbröl et al., 2022).



## 3 Methodology and retrieval techniques

Our intent is to assess the consistency of products retrieved from the FINESSE radiances with equivalent information derived from the available in-situ airborne and ancillary ground-based observations. To this end, three retrieval techniques have been applied to the available datasets: (Section 3.1) the first to the FINESSE spectral radiances to retrieve the cirrus effective diameter ($D_{ei}$) and visible optical depth (OD), together with the atmospheric profiles of water vapour and temperature and the ozone total column; (Section 3.2) the second to the ceilometer backscattering signal to retrieve the cirrus extinction profiles and then the OD; (Section 3.3) the third to the HATPRO radiometer measurements to retrieve the vertical profiles of water vapour and temperature.

### 3.1 Retrieval of atmospheric and cloud parameters from FINESSE spectral radiances

The FINESSE analysis was performed using the Simultaneous Atmospheric and Clouds Retrieval (SACR) code (Di Natale et al., 2020). To adapt the code to simulate the FINESSE radiances we integrated the instrument line shape described in Murray et al. (2024) within the SACR Forward Model (FM) so that the high-resolution spectral radiances generated by the FM were correctly convolved to mimic the spectrum measured by FINESSE. Unless otherwise stated the retrievals exploit the spectral range 400-1500 cm$^{-1}$.

The FM simulations were performed on a vertical grid fixed between 0.37 (ground level) and 70 km, to account for the radiative contribution of stratospheric ozone. To resolve detail within the boundary layer, a finer vertical resolution of 0.02 km was chosen close to the surface, reducing to 1 km at the highest altitudes. The cirrus cloud is modeled by fixing the position between 6-7 km and as a multilayer with an internal vertical resolution of 0.1 km. The initial guess and a-priori for temperature and water vapour profiles and ozone total column used in the retrieval, were built by perturbing the corresponding profiles provided by ECMWF ReAnalysis-5 (ERA5) (Hersbach et al., 2020) database. Water vapour was perturbed by assuming the vertical distribution of uncertainties used for The Michelson Interferometer for Passive Atmospheric Sounding (MIPAS) analysis (Remedios et al., 2007)); temperature and ozone were perturbed, respectively, by 1% and 10% of the corresponding profiles. The concentrations of other minor gases were provided by the Initial Guess-2 (IG2) dataset (Remedios et al., 2007), with $CO_2$ scaled to the value provided by the FINESSE Vaisala probe. A-priori estimates for $D_{ei}$ and OD were set equal to 80 $\mu$m and 0.5, respectively (that are values bit higher than those expected for thin cirrus clouds) assuming uncorrelated errors as large as 100%. These values were also set as initial guesses in the analysis.

In Table 2 are reported the retrieved parameters with FINESSE, the initial guess and a priori values used in the retrieval procedure, the a priori errors considered, the alternative source/measurements providing the parameters with which we compared our results or we fixed in the radiative transfer simulations.

### 3.1.1 Simulation of cirrus cloud spectral radiance

The SACR FM combines spectral gaseous ODs generated by the Line-By-Line Radiative Transfer Model (LBLRTM v12.15) (Clough et al., 2005) with a specific code based on the two stream $\delta$-Eddington approximation which can account for multiple





**Table 2.** Parameters retrieved from FINESSE spectra, those derived from alternative simultaneous and co-located source/measurements used to validate the retrieval products and the unretrieved parameters used to simulate the radiative transfer.

| FINESSE retrieved parameter | initial guess/a priori | a priori error | alternative source | not retrieved parameter |
| --- | --- | --- | --- | --- |
| $D_{ei}$ | 80 $\mu$m | 100% | INCAS KA | habit type, $T_{cld}$ |
| OD | 0.5 | 100 % | CEILOMETER | CTH, CBH |
| T profile | ERA5 perturbed of 1% | 1.5% | ERA5 | - |
| T @ ground | ERA5 perturbed of 1% | 1.5% | VAISALA | - |
| WV profile | ERA5 perturbed following MIPAS analysis | 100% | ERA5 | - |
| WV @ ground | ERA5 perturbed following MIPAS analysis | 100% | VAISALA | - |
| $O_3$ total column | ERA5 perturbed of 10% | 100% | ERA5 | - |
| - | - | - | IG2 + VAISALA | $CO_2$ profile |
| - | - | - | IG2 | $N_2O$, $CH_4$, CO profiles |

scattering from particles. SACR is designed such that it can ingest the single scattering properties of spherical water droplets alongside those of ice crystals with different habits. Ice crystal properties are taken from the Yang et al. (2015) databases that tabulate the extinction/absorption efficiencies ($Q_{a,ei\nu}$), the single scattering albedos ($\omega_\nu$) and the asymmetry factors ($g_\nu$)

together with the volumes ($V_h$) and projected areas ($A_h$) of nine crystal habits (the subscript h denotes the habit), namely solid/hollow bullet rosettes (SBR/HBR), solid/hollow columns (SCL/HCL), plates (PL), droxtals (DX), spheroids (SPH), aggregates (AGG) (10-plates and 8-columns), as a function of the crystal maximum length (L) in the range 2–10000 $\mu$m and wavenumber ($\nu$) in the range 100–5000 cm$^{-1}$. To model the cirrus SACR uses the averages of the single scattering properties integrated over assumed particle size (n(L)) and habit ($f_h$(L)) distributions tabulated in the form of look-up tables (LUTs) as a

function of $D_{ei}$. $D_{ei}$ is defined as follows (Yang et al., 2005):

$$D_{ei} = \frac{3}{2} \frac{\sum_{h=1}^{N} \int_{L_{min}}^{L_{max}} f_h(L) V_h(L) n(L) dL}{\sum_{h=1}^{N} \int_{L_{min}}^{L_{max}} f_h(L) A_h(L) n(L) dL} \tag{1}$$

$f_h$ is defined such that $\sum_{h=1}^{N} f_h = 1$ at each particle length $L$ and $N$ is the total number of habits. The averages of the single scattering properties $\langle Q_{a,ei} \rangle_\nu$, $\langle g_i \rangle_\nu$ and $\langle \omega_i \rangle_\nu$ are given by:

$$\langle Q_{a,ei} \rangle_\nu = \frac{\sum_{h=1}^{N} \int_{L_{min}}^{L_{max}} f_h(L) Q_{a,ei,\nu}(L)(L) A(L) n(L) dL}{\sum_{h=1}^{N} \int_{L_{min}}^{L_{max}} f_h(L) A(L) n(L) dL} \tag{2}$$

$$\langle g_i \rangle_\nu = \frac{\sum_{h=1}^{N} \int_{L_{min}}^{L_{max}} f_h(L) g_i(L) Q_{si,\nu}(L) A(L) n(L) dL}{\sum_{h=1}^{N} \int_{L_{min}}^{L_{max}} Q_{si,\nu}(L) f_h(L) A(L) n(L) dL} \tag{3}$$





$$\langle \omega_{\mathrm{i}} \rangle_\nu = 1 - \frac{\langle Q_{\mathrm{ai}} \rangle_\nu}{\langle Q_{\mathrm{ei}} \rangle_\nu} \tag{4}$$

where $Q_{\mathrm{si},\nu} = Q_{\mathrm{ei},\nu} - Q_{\mathrm{ai},\nu}$ is the scattering efficiency and $L_{min} = 2\ \mu$m and $L_{max} = 10000\ \mu$m represent the limits of maximum lengths. Finally, the expression of the OD for a generic wavenumber ($\mathrm{OD}_\nu$) is given by:

$$\mathrm{OD}_\nu = \frac{3 \cdot \mathrm{IWP}}{D_{\mathrm{ei}} \rho_i} \frac{\langle Q_{\mathrm{ei}} \rangle_\nu}{2} = \mathrm{OD} \frac{\langle Q_{\mathrm{ei}} \rangle_\nu}{2} \tag{5}$$

with IWP the integrated water path and $\rho_i = 917\ \mathrm{kg\ m}^{-3}$ the ice density.

### 3.1.2    Building the look-up tables of single-scattering average properties

To build the LUTs of the cirrus clouds average properties as a function of the effective diameter and wavenumber we applied Eqs. (1)–(4) and assumed a-priori a monomodal $\Gamma$-like PSD (Platnick et al., 2017; Turner, 2005) that varies analytically with the modal parameter $L_m$:

$$n(L) = N_o L^\mu e^{-(3+\mu)\frac{L}{L_m}} \tag{6}$$

where the quantity $L_m/(3+\mu)$ represents the mode of the distribution. In this way, we obtained a set of $\Gamma$ PSDs (each corresponding to a $D_{ei}$) to use in Eqs. (1)–(4) simply by defining a vector of continuous values of $L_m$ between 2 and 10000 $\mu$m . The intercept parameter $N_o$ does not need to be evaluated since it appears in both the numerator and denominator in Eqs. (1)–(4) while the dispersion coefficient $\mu$, that is temperature dependent, must be set. This coefficient was assessed by fitting 225    the average of the INCAS KA measured PSDs (see section 2.1) with the function described by Eq. 7; we found a best fit value of $\mu = 2$ and we used it in Eqs. (1)–(4) to create the LUTs for four habit distributions described below.

Fig. 7 shows that the ice crystals sampled on the 17th February flight were predominantly of bullet rosette morphology. However, there are also a minority of crystals, below $\sim$50 $\mu$m of maximum length, which are difficult to define from the photographs. For these crystals the aspect ratio tends to be close to 1 and we make the assumption that they are plate-like, consistent with previous work performed over the Antarctic Plateau (Del Guasta, 2022). With this assumption we define the 230    measured habit distribution as follows:

$$f_h(L) = \begin{cases} \mathrm{BR} & L > 50\ \mu m \\ \mathrm{PL} & L \leq 50\ \mu m \end{cases} \tag{7}$$

where BR denotes generic bullet rosettes. The habit distribution in Eq. (7) is used in Eqs. (1)–(4) together with the PSDs to build the LUTs of the cirrus cloud average properties as a function of the effective diameter and wavenumber. To test the 235    impact on the retrieval results of mixing other habits we also considered four habit distributions: two by assuming single HBR



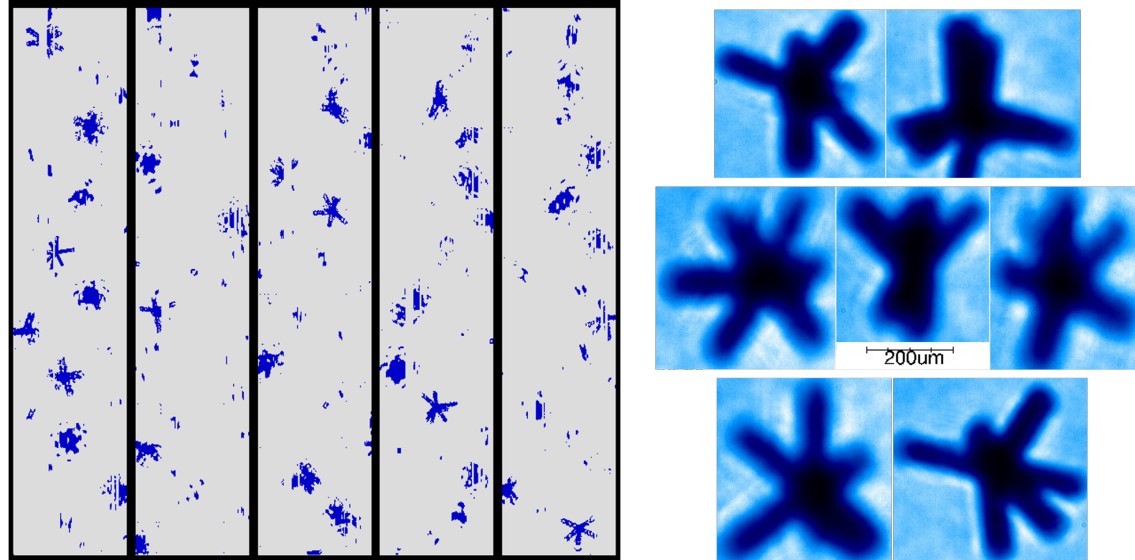

**Figure 7.** Examples of the ice crystals observed inside the cirrus layer by the INCAS KA on 17th February 2023 at 8:20 UTC. Images from the 10 $\mu$m-resolution channel of the 2D-S (left) where the width of each strip is 1280 $\mu$m and the CPI (right) are shown

and SBR, denoted as f1h and f2h, and the others, $f_{3h}$ and $f_{4h}$, by mixing them homogeneously and also adding solid columns. Considering Eq. (7), these habit distributions can be summarized as follows:

$$f_{1h}(L) = \begin{cases} \text{HBR} & L > 50 \ \mu m \\ \text{PL} & L \leq 50 \ \mu m \end{cases} \tag{8}$$

$$f_{2h}(L) = \begin{cases} \text{SBR} & L > 50 \ \mu m \\ \text{PL} & L \leq 50 \ \mu m \end{cases} \tag{9}$$

$$f_{3h}(L) = \begin{cases} \frac{1}{2}\text{HBR} + \frac{1}{2}\text{SBR} & L > 50 \ \mu m \\ \text{PL} & L \leq 50 \ \mu m \end{cases} \tag{10}$$

$$f_{4h}(L) = \begin{cases} \frac{1}{3}\text{HBR} + \frac{1}{3}\text{SBR} + \frac{1}{3}\text{SCL} & L > 50 \ \mu m \\ \text{PL} & L \leq 50 \ \mu m \end{cases} \tag{11}$$

### 3.1.3 Retrieval algorithm

The SACR retrieval algorithm is based on the optimal estimation (OE) approach and the Levenberg-Marquardt iterative formula Marquardt (1963); Rodgers (2000). Initially we use FINESSE observations over the spectral range 400–1500 cm$^{-1}$ to perform the retrieval. The atmospheric state vector in the presence of cirrus clouds used by SACR is:





$$\mathbf{x} = (D_{ei}, OD, \mathbf{U}, \mathbf{T}, O_3, \beta) \tag{12}$$

where $\mathbf{U}$ and $\mathbf{T}$ denote the vectors of the selected retrieval levels for the water vapour and temperature profiles, $O_3$ indicates the ozone total column, and $\beta$ a frequency stretch (usually of the order of $10^{-6}$) which takes into account the shift due to the finite internal aperture ($\Omega$) given by $\nu(1 - \frac{\Omega}{4\pi})$ (Davis et al., 2001), and the effect due to possible instability of the reference

laser (Palchetti et al., 2016). The levels in $\mathbf{U}$ and $\mathbf{T}$ were set to 0.37 (ground level), 0.60, 0.80, 1.00, 1.40, 1.80, 2.0, 3.0, 4.0 km for water vapour and 0.37, 0.39 and 0.80 km for temperature, respectively. In each case, above the highest level the profile is rescaled keeping the shape of the original profile. The levels were chosen from a preliminary sensitivity study and are designed to retrieve profile structure without over-constraining the solution with a-priori information, so avoiding non-physical fluctuations.

The algorithm finds the solution as that which minimises the cost function:

$$\chi^2 = (\mathbf{y} - \mathbf{F}(\mathbf{x}))^T \mathbf{S}_y^{-1}(\mathbf{y} - \mathbf{F}(\mathbf{x})) + (\mathbf{x} - \mathbf{x}_a)^T \mathbf{S}_a^{-1}(\mathbf{x} - \mathbf{x}_a) \tag{13}$$

where $\mathbf{y}$, $\mathbf{F}$ and $\mathbf{x}_a$ denote the vector of the FINESSE measurements, the forward model and the vector of the a-priori information, respectively. The a-priori correlation lengths for water vapour and temperature profiles were set to 0 and 2 km, respectively, to avoid over-constraining the retrieval and because the temperature shows a smooth linear behavior with altitude.

However, the correlation between the two first temperature levels was set to 0 because the very first layers can be locally affected by the presence of nearby thermal sources.

The cost function is minimized with the LM iterative formula as follows (Rodgers, 2000):

$$\mathbf{x}_{i+1} = \mathbf{x}_i + [\mathbf{K}_i^T \mathbf{S}_y^{-1} \mathbf{K}_i + \gamma_i \mathbf{D}_i + \mathbf{S}_a^{-1}]^{-1} [\mathbf{K}_i^T \mathbf{S}_y^{-1}(\mathbf{y} - \mathbf{F}(\mathbf{x}_i)) - \mathbf{S}_a^{-1}(\mathbf{x}_i - \mathbf{x}_a)] \tag{14}$$

with $\mathbf{S}_y$ and $\mathbf{S}_a$ the Variance Covariance Matrices (VCMs) of the measurements and the a-priori information; $\gamma_i$ is the

damping factor at the $i$-h iteration, $\mathbf{K}_i$ represents the Jacobian matrix of $\mathbf{F}$ and $\mathbf{D}_i$ is a diagonal matrix as described in Di Natale et al. (2020). The FINESSE measurement noise is considered uncorrelated and given by the noise equivalent spectral ratio (NESR) calculated as discussed in Murray et al. (2024). Convergence is reached when the variations in $\chi^2$ are less than 1‰. The retrieval errors of the parameters are contained in the VCM given by the relation (Rodgers, 2000):

$$\mathbf{S}_x = (\mathbf{K}^T \mathbf{S}_y^{-1} \mathbf{K} + \mathbf{S}_a^{-1})^{-1} \tag{15}$$

where $\mathbf{K}$ is the Jacobian matrix at the last iteration.



### 3.1.4 Accounting for calibration uncertainty

We chose to study a-posteriori the effect of the systematic error due to FINESSE calibration uncertainty, since including the spectral correlation effect in the VCM of measurements could lead, based on experience, to stuck the retrieval algorithm in local minima and so to incorrect results when the spectrum is in presence of clouds and all spectral channels are considered in the procedure. Our initial retrievals from FINESSE assume no calibration uncertainty on the observed radiances. We then repeat the retrievals systematically increasing and reducing the radiances with their calibration uncertainties (Fig. 8). The average between the maximum and the minimum differences of the retrieval products with respect to those obtained from the original retrieval are assumed to be the contribution of the calibration error. Total retrieval errors are then calculated as the square root of the quadratic sum of the retrieval error (equation 15) and the calibration error. These are the values reported in Section 4.

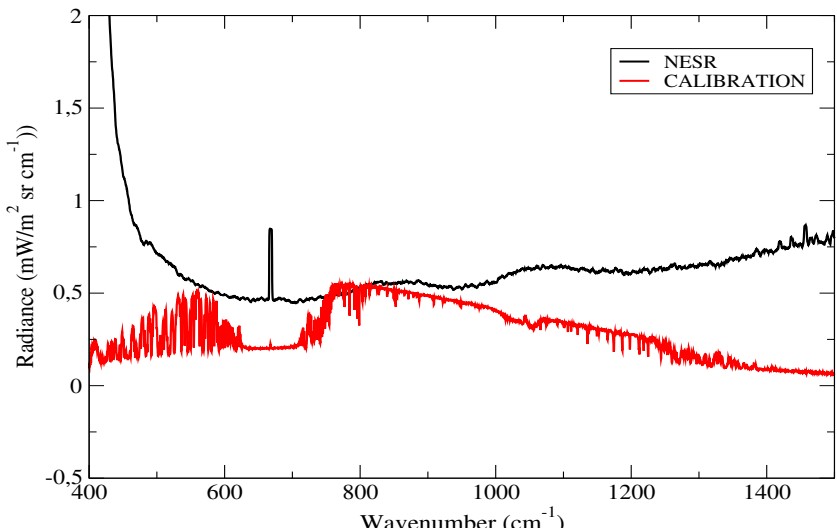

**Figure 8.** Comparison of the FINESSE Noise Equivalent Spectral Ratio (NESR, black curve) and the calibration systematic error (red curve).

### 3.2 Klett method for retrieving cirrus clouds optical depth from ceilometer backscattering signal

The cirrus OD can also be retrieved from the ceilometer backscattering signal, that we call $P$, as a function of the height $z$ by using the Klett inversion method (Klett, 1981, 1985). Since this problem involves only one equation and two unknowns, namely the backscatter ($\beta_e$) and the extinction ($\sigma_e$), the relationship that binds them must be assumed. A common solution is to assume a power-law relationship with a generic exponent $k$, called the backscatter-extinction coefficient ratio, as follows:

$$\beta_e(z) = C \cdot \sigma_e(z)^k \tag{16}$$

where $C$ is a constant and $k$ depends on the laser frequency, the aerosol composition and the inhomogeneity of the layers, particularly the variability of the aerosol PSD, the particles shape and the mixing with the air, and is affected by multiple-



scattering effect (Del Guasta et al., 1993; Elouragini, 1995). $k$ generally ranges between 0.67–1.00 (Klett, 1981; Elouragini, 1995), due to the complexity of particle shapes and their inhomogeneous dimensions and distributions: preliminary tests by varying k suggested a value of k=1 was most appropriate for our analysis here. Using the relationship of Eq. (16), the Klett's lidar solution for the cloud extinction profile ($\sigma_c(z)$) is obtained as a function of the height $z$ as follows (Klett, 1981):

$$\sigma_c(z) = \frac{\exp\left[\frac{S(z)-S_r}{k}\right]}{\sigma_r^{-1} + \frac{2}{k}\int\limits_z^{z_r}\exp\left[\frac{S(z')-S_r}{k}\right]dz'} - \sigma_m(z) \tag{17}$$

where $S(z) = \ln(P(z)z^2)$ represents the logarithmic range-corrected signal (LRCS), $\sigma_m(z)$ is the molecular extinction due to the gas contribution derived from the atmospheric profiles, $S_r$ and $\sigma_r$ denote the LRCS and the extinction at a reference height $z_r$ that must be fixed above the CTH, where the backscattering is completely molecular in origin. Because of the noisiness of the signal above the CTH and the sensitivity of the solution to $\sigma_r$, a signal smoothing process, such as a moving average, is necessary to make the solution more stable. From the extinction profile the optical depth is obtained as the integral over the cloud geometrical thickness:

$$OD = \int\limits_{CBH}^{CTH} \sigma_c(z)dz \tag{18}$$

Since the solution is sensitive to $\sigma_r$ we repeated the retrieval procedure, varying the choice of $z_r$ above the CTH. We used 10 heights from 0.1 km to 1 km in steps of 0.1 km and report the average value as the ceilometer derived OD. This approach also allows to evaluate the retrieval error for each scenario as the standard deviation of the retrieved OD divided by $\sqrt{10}$, the number of heights chosen.

### 3.3 Retrieval of temperature and water vapour vertical profiles from HATPRO measurements

The HATPRO temperature and humidity profiles are based on statistical retrievals (Löhnert and Crewell, 2003). Using a twenty year ERA5 dataset for Andenes a quadratic regression model was trained relating atmospheric profiles and simulated brightness temperatures. The "pyMakeRetrieval" software package was used to create temperature and humidity retrievals (Foth, 2024).

### 3.4 Calculation of the effective diameters of ice crystals inside cirrus clouds from INCAS measurements

The INCAS KA measurements offer a check of the retrieval of $D_{ei}$ from the FINESSE spectral radiances by directly substituting the observed PSDs measured during the flight into Eq. (1). As noted in section 3.1.1, the calculation depends on the specific habit distribution assumed. We repeat calculations for the four habit distributions described in Eqs. (1)–(4) for observations taken by the INCAS KA between 08:19 and 09:00 UTC, neglecting periods at the highest 'cloud' altitudes, in the 20 m above 6.77 km where tenuous and highly variable signals were reported. The slightly longer time-span in comparison to the selected FINESSE measurements is to enable sufficient statistics to be generated. Fig. 9 shows the effective diameters calculated using INCAS KA PSDs as a function of the height inside the cirrus cloud.





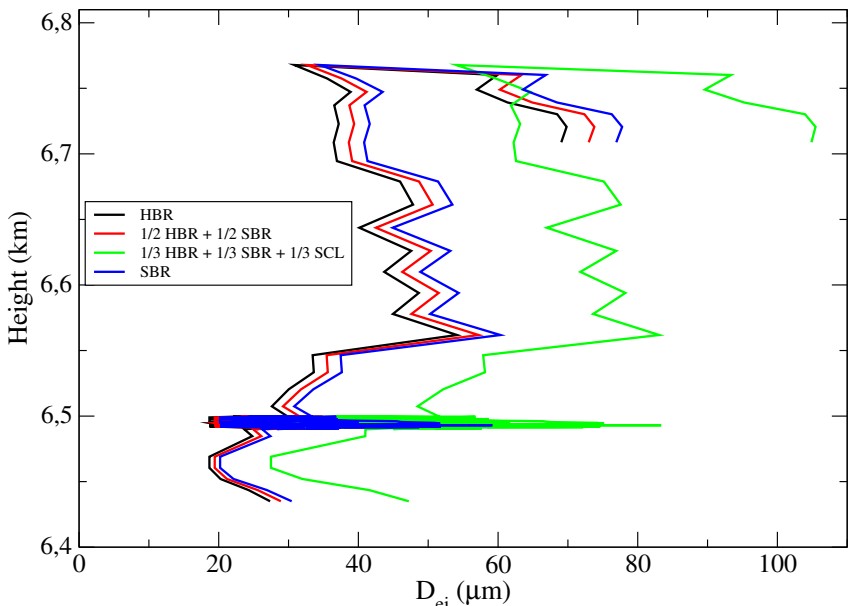

**Figure 9.** Vertical distribution of the derived ice crystal effective diameters calculated from the INCAS KA observed ice crystal PSDs and using the Yang et al. (2013) single scattering properties databases and the habit distribution given by Eqs. (1)–(4).

To assess the accuracy of the PSDs measured by the INCAS KA, and estimate the systematic error that affects the effective diameter due to the bin sampling of the PSD, we took into account that the measured PSD associates the same number of ice particles within a specific crystal length bin equal to 50 $\mu$m , starting from a minimum length of 25 $\mu$m . We considered three PSDs normalized to the respective maxima and corresponding to three different effective diameters, the smallest one equal to

25 $\mu$m , the middle one equal to 49 $\mu$m and the largest one equal to 73 $\mu$m . Since we want to assess how much the bin sampling limits the accuracy, we need to find the analytic functions that reproduce the data distribution as closely as possible. We fitted the first PSD with the mono-modal $\Gamma$-function in Eq. (6) while the second and third were fitted with a bi-modal $\Gamma$-function:

$$n(L) = N_1 L^{\mu_1} e^{-(3+\mu_1)\frac{L}{L_{m1}}} + N_2 L^{\mu_2} e^{-(3+\mu_2)\frac{L}{L_{m2}}} \tag{19}$$

obtaining the best fit curves shown in Fig. 10. For each PSD we calculated the difference in the value of the effective

diameters obtained by using the measured PSD and the corresponding fitted curve; then we selected the maximum difference for the 3 cases, which was approximately 5 $\mu$m . This corresponds to the systematic error induced by the bin approximation implicit in the INCAS KA data. Finally, we added the statistical standard deviation on the average effective diameters obtained from the selected INCAS PSDs, equal to 1 $\mu$m . The two errors, systematic and statistical, were summed in quadrature to give a combined uncertainty of $\sim$5 $\mu$m on the effective diameters derived from the INCAS measurements.





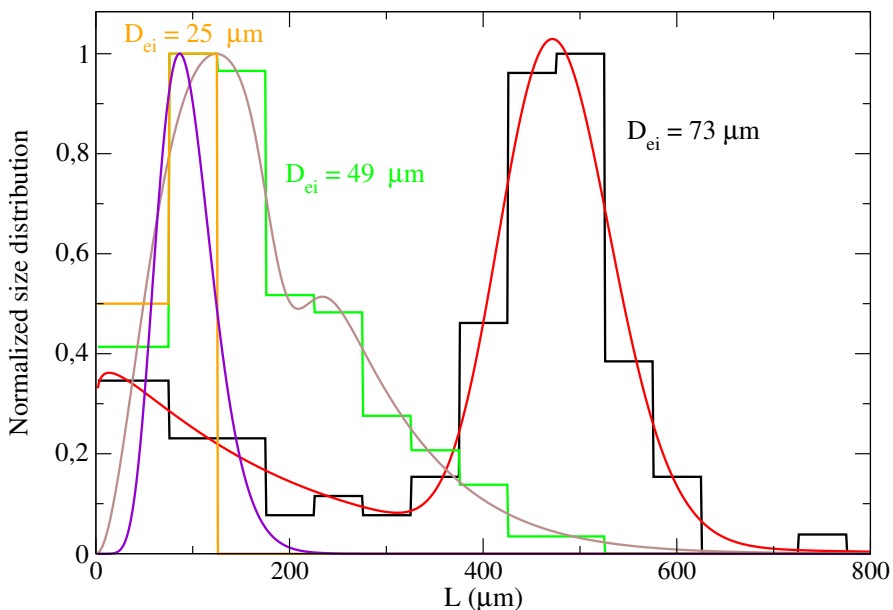

**Figure 10.** Three examples of PSDs normalized to the respective maxima measured by the INCAS KA (black, green and orange histograms) with the $D_{ei}$ calculated assuming the habit distribution in Eq. (7) and the fitting functions (red, brown and violet curves)

## 4 Results

### 4.1 Water vapour and temperature profiles

Fig. 11 shows a comparison of the retrieved profiles of temperature and water vapour from the FINESSE observations at 08:25:29 UTC with co-located ERA5 profiles (Hersbach et al., 2020). For a fairer comparison, the ERA5 data are convolved with the appropriate averaging kernels. In general, relative to the first guess, the information from FINESSE tends to moisten the lowest 1 km of the atmosphere and dry the layer between 2-3 km. This behavior is consistently seen for all 11 cases identified in Table 1. Although we do not possess a 'true' water vapour profile, we can see that the uncertainties on the retrieval typically overlap the ERA5 estimates. For temperature, the retrievals at the lowest levels are shifted warmer and closer to the ERA5 values. The in-cloud temperature structure recorded by the INCAS KA temperature probe also shows excellent overlap with ERA5, giving additional confidence in the latter as a realistic representation of the atmospheric state. Moreover, even though the upper part of the retrieved temperature profile is only rescaled with respect to the highest retrieval level, keeping the shape of the initial guess profile, as described in section 3.1.3, the matching with the INCAS KA data provides a very good indication that the retrieval is consistent and the atmospheric state fairly characterised and represented in the simulations. The average difference between the retrieved and INCAS KA temperature profile is 0.37 K and all differences vary inside the interval -1 to 1 K.



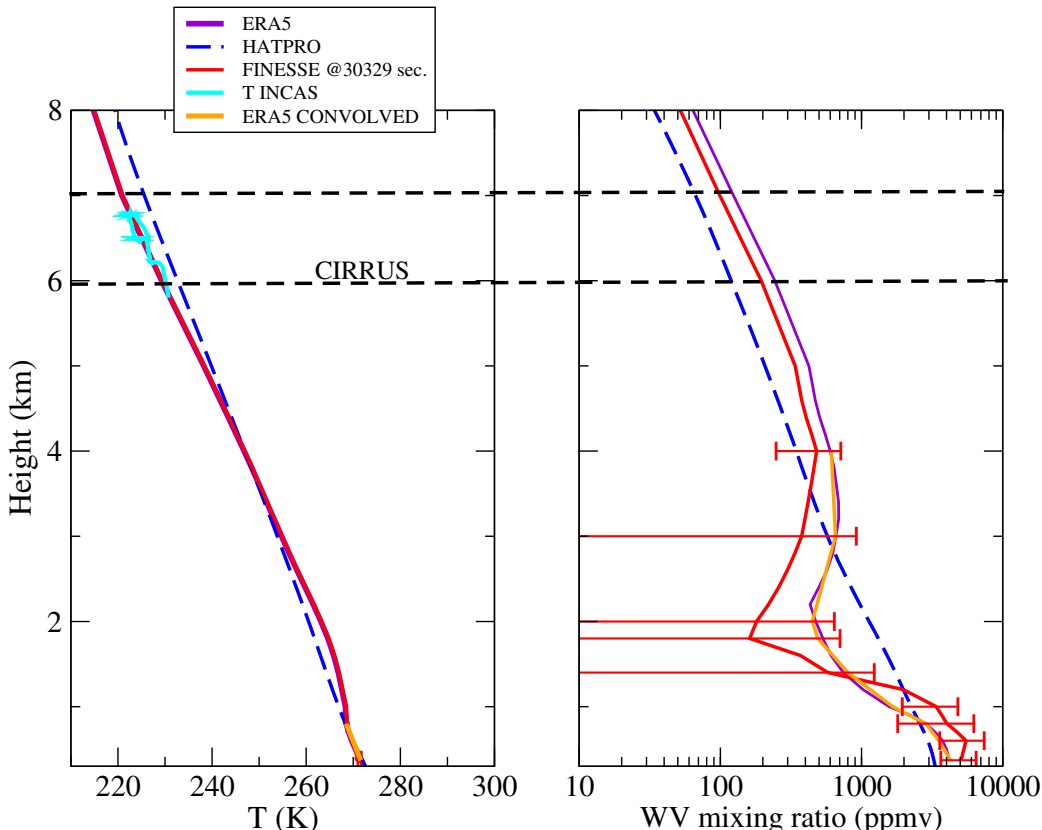

**Figure 11.** Left panel: comparison of the retrieved temperature profile from FINESSE spectrum (red solid line) at 30329 sec (08:25:29 UTC, scenario 4) since midnight of 17 February 2023 with those provided by ERA5 (violet solid line), ERA5 convolved with the averaging kernels at the last iteration (orange solid line), INCAS KA (cyan solid line), and HATPRO radiometer (dashed blue line); in brown the initial guess/a-priori is also reported. Right panel: same as the left panel but for water vapour profiles. The cirrus location is denoted by the dashed horizontal lines.

For completeness Fig. 11 also includes the HATPRO retrieved profiles. The temperature retrievals tend to show a cold bias relative to ERA5 between 1-3 km, switching to a more pronounced warm bias above 5 km. As might be expected in light of previous studies, the water vapour profile is too smooth, and not able to capture the vertical structure, particularly the dry layer between 1-3 km.

**4.2    Near-surface temperature and water vapour mixing ratios; total precipitable water and total column ozone**

Fig. 12 provides a comparison between water vapour concentrations and temperature measured by the FINESSE Vaisala probes, the precipitable water vapour from the HATPRO and corresponding information derived from the FINESSE spectra and ERA5 data. A comparison of the ERA5 total column ozone with the equivalent FINESSE retrievals is also provided.





Considering near-surface temperature, FINESSE retrievals tend to sit between the values recorded by the Vaisala sensor and that given by ERA5 (Fig. 12, top panel). For the first five scenarios, when the overhead cirrus was optically thicker (Fig. 13),

the FINESSE retrievals tend to show better agreement with the ERA5 value. Meanwhile, with thinner cirrus overhead they typically show an improved agreement with the Vaisala data (scenario 7-11). In all cases the difference between the FINESSE products and Vaisala data is lower than 0.7 K. The near-surface water vapour volume mixing ratios (VMR) obtained from the FINESSE retrievals show excellent agreement with the Vaisala probe data (Fig. 12, second panel). In contrast, ERA5 appears to underestimate the VMR for all scenarios, with differences reaching up to 1000 ppmv. The corresponding uncertainties for

ERA5 quantities are assessed in Hersbach et al. (2020); Dragani et al. (2017); Bell et al. (2021).

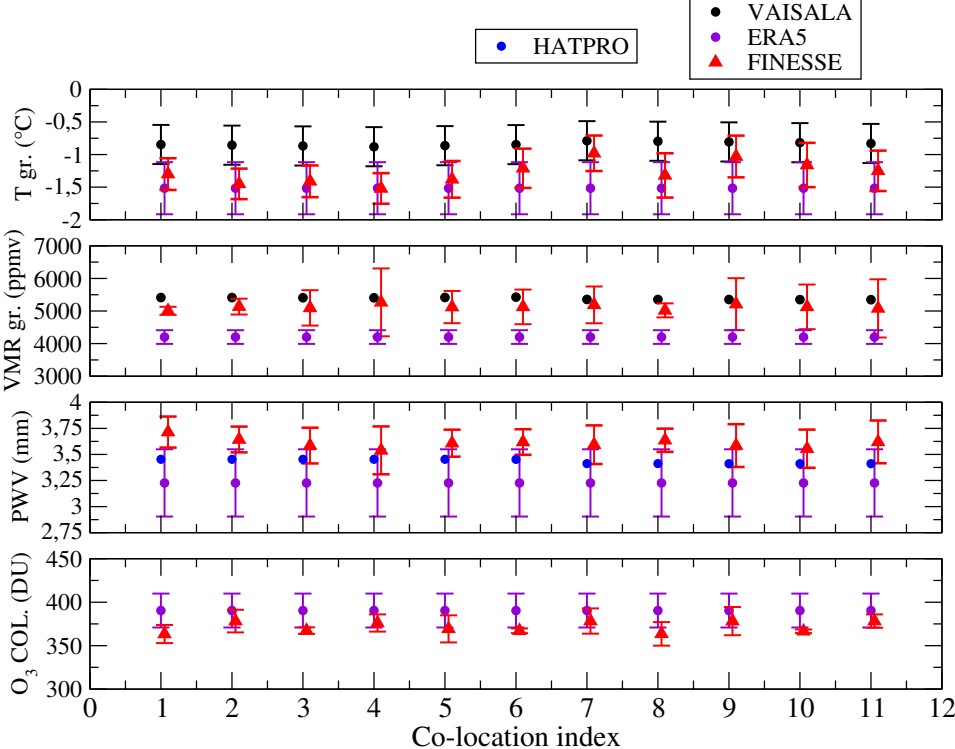

**Figure 12.** Top panel: comparison of the temperature at ground retrieved from FINESSE spectra (red triangles), measured by the Vaisala probe (black dots) and ERA5 (violet dots) for each scenario. Second panel: same comparison as the temperature but for water vapour mixing ratio (VMR). Third panel: comparison of precipitable water vapour (PWV), retrieved from FINESSE (red triangles), from integrated HATPRO retrievals (blue dots) and from ERA5 (violet dots). Bottom panel: comparison of the ozone total column retrieved from FINESSE and provided by ERA5.

The tendency for ERA5 to underestimate near surface water vapour relative to FINESSE is also reflected in comparisons of precipitable water vapour (PWV) (Fig. 12, third row), with a low bias which is on the order of 10% in this relatively dry environment. HATPRO PWV retrievals show values which sit between the FINESSE and ERA5 estimates but are typically





closer to the former. Finally, the lowest panel of Fig. 12 indicates that the total column ozone retrieved from FINESSE is close
to the values obtained from ERA5 ozone profiles even though a bias of about 20 DU is present.

## 4.3 Cloud parameters

### 4.3.1 Optical Depths

The cirrus OD was retrieved independently from FINESSE spectra and the ceilometer backscattering signal as described
in sections 3.1 and 3.2 assuming HBRs. The comparison between the two retrievals is reported in Figure 13. Overall, the
values provided by FINESSE retrieval and those derived from ceilometer show agreement within their respective retrieval
uncertainties. The ceilometer ODs show a significantly higher uncertainty because of the noisiness of the ceilometer signal
above the CTH, which propagates through the Klett inversion algorithm.

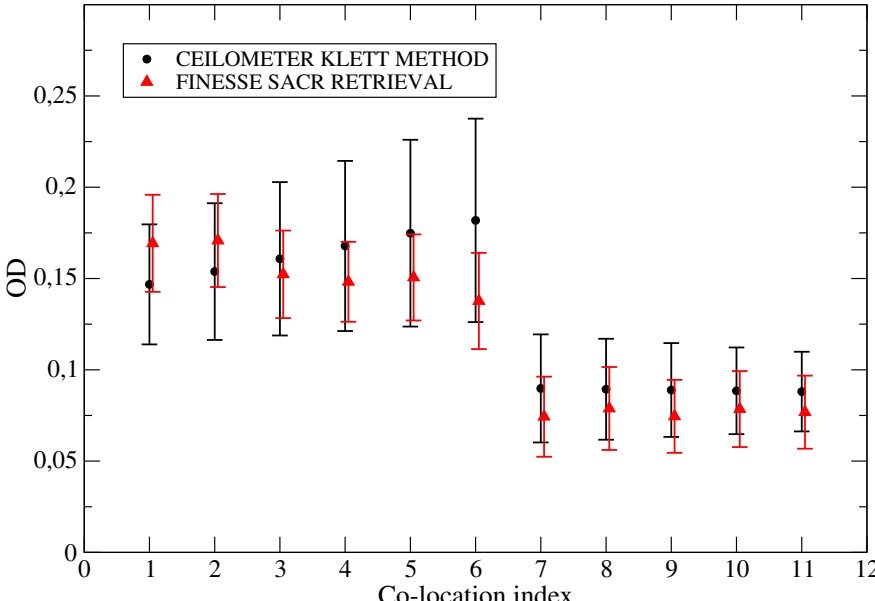

**Figure 13.** Comparison of the OD retrieved from FINESSE spectra (red triangles) and those retrieved from the ceilometer signal (black dots)
for each scenario.

Despite these uncertainties Fig. 13 clearly shows a distinct thinning of the cirrus between the first six scenarios (centered
around 08:25:30 UTC) and the second five (centered around 08:37:45 UTC). The mean OD drops from 0.17 to 0.08, in
particular we obtained an average value retrieved from radiometric measurements equal to (0.130±0.004), with the associated
error calculated as standard deviation of the mean, compared with the average value obtained from ceilometer measurements
equal to (0.120±0.004). This indicates an excellent accordance between the retrievals from the different measurements.

The effect of this reduction in OD is clearly manifested in the FINESSE spectra as illustrated in Fig. 14, where the black
curve represents the spectrum taken at 08:25:23 UTC and the red curve shows the spectrum measured at 08:37:49 UTC.




For these specific spectra the retrieved OD drops from 0.169 to 0.077. The change in OD most obviously manifested in the atmospheric window with a reduction in both the radiance level and the spectral gradient between 750–1000 cm$^{-1}$.

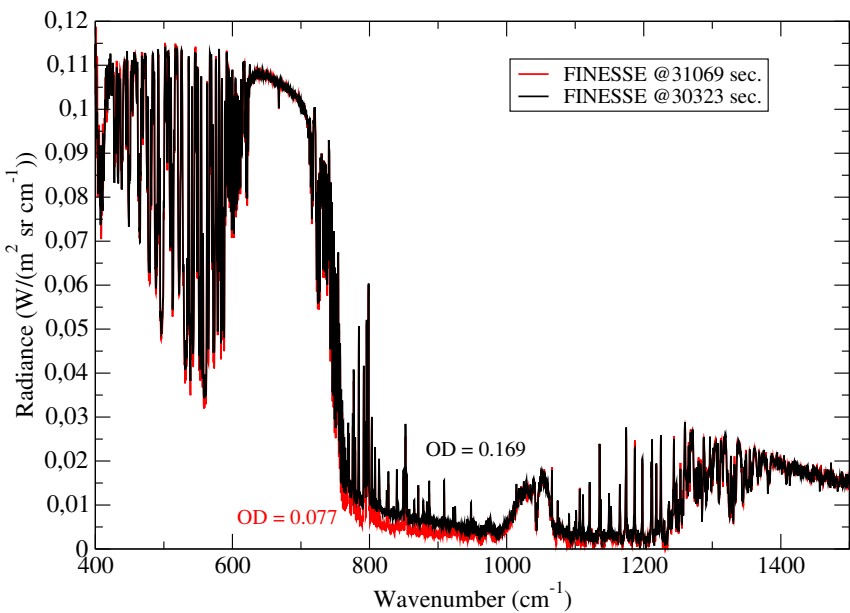

**Figure 14.** Spectra measured by FINESSE in the presence of cirrus at 30323 and 31069 sec. since midnight (08:25:23 and 08:37:49 UTC) on 17 February 2023. The associated retrieved cirrus optical depths are also provided.

### 4.3.2 Cirrus effective diameter

Effective diameters retrieved from FINESSE were compared with those obtained from the PSDs provided by the aircraft sensors inside the cirrus. These values were calculated by assuming the four habit distributions introduced in section 3.1.1 and

are shown as a function of the altitude reached by the INCAS KA in Fig. 15.

The violet solid lines denote the mean values obtained by averaging the $D_{ei}$ calculated at each altitude using the INCAS KA PSDs. The associated error is calculated as described in section 9. Each panel corresponds to the results obtained by assuming a specific habit distribution discussed in section 3.1.2. The red triangles indicate the values retrieved from FINESSE. The assumption of HBRs, either in isolation or mixed with SBRs (top and second panel) clearly show better agreement with

the aircraft data than the other assumptions of pure SBRs or a mixture with SCLs (third and bottom panel). In particular, the retrievals obtained when assuming only HBRs show a very good agreement with the aircraft data, with almost all retrieved $D_{ei}$ consistent within uncertainties (top panel). Indeed, over the 11 cases considered, the average retrieved value from the radiometric measurements is (26.3±0.5) $\mu$m assuming HBRs which shows good consistency with the average value derived from the in-situ measurements of (31.5±5.0) $\mu$m . The results clearly indicate the strong sensitivity of the FINESSE measurements

(and hence retrievals) to the cirrus crystal habit.





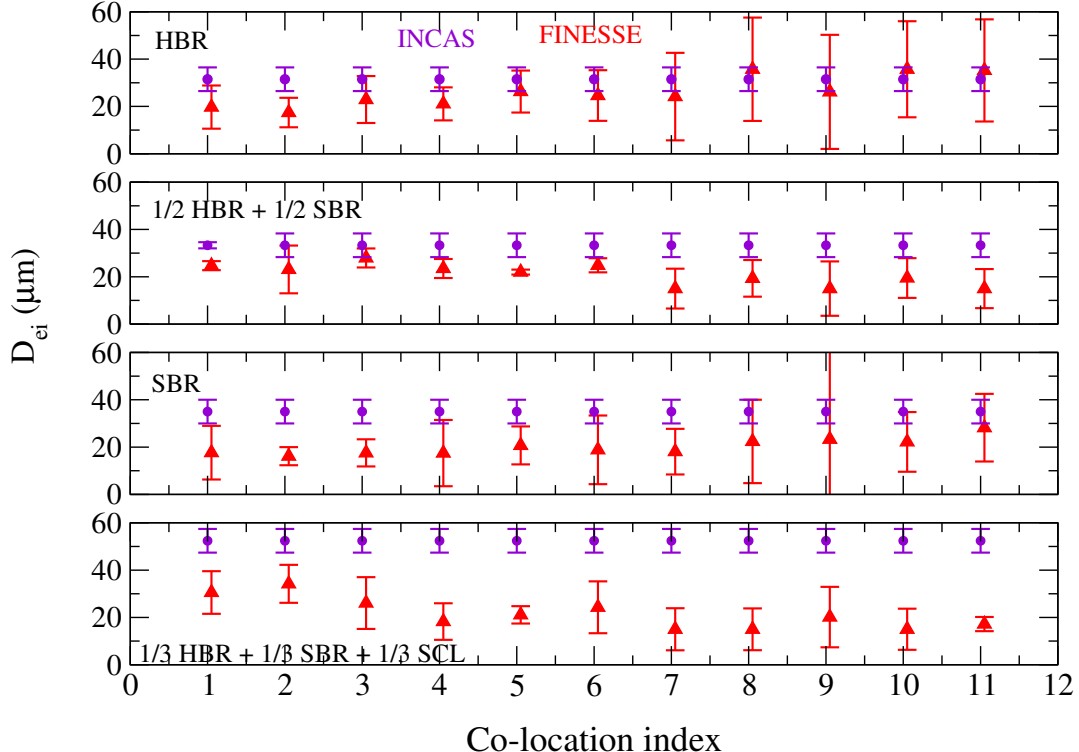

**Figure 15.** Top panel: comparison of the effective diameters calculated from INCAS KA data (violet dots) between 08:19–09:00 UTC on 17 February 2023 and those retrieved from FINESSE (red triangles) between 08:25–08:38 UTC of the same day by assuming the habit distribution $f_{1h}$ in Eq. (8). Second panel: same comparison but assuming $f_{3h}$ from Eq. (10). Third panel: same comparison but assuming $f_{2h}$ from Eq. (9). Bottom panel: same comparison but assuming $f_{4h}$ from Eq. (11). For brevity, the caption labels do not explicitly state the fraction of plates considered for crystal lengths below 50 $\mu$m as discussed in section 3.1.1

## 5 Influence of the FIR

In this section we show the importance of information contained within the FIR portion of the spectrum in constraining the cloud property retrievals. Fig. 16 shows the difference in simulated downwelling radiances between the 'best-fit' HBR case and the other three habit choices for cirrus with a $D_{ei}$ of 30 $\mu$m and OD of 0.2. The highest sensitivity to crystal shape is seen at wavelengths within the FIR, between 470-600 cm$^{-1}$. This band is delimited by orange dashed lines in Fig. 16. Moreover, for all three sets of residuals, the differences only exceed the FINESSE NESR between approximately 500-580 cm$^{-1}$, implying that the radiance fit over this range plays a critical role in determining the retrieved size and optical depth for each habit choice. In essence it is these FIR radiances that allow discrimination between the habits, with the selection of HBR agreeing with the INCAS KA 2D-S and CPI imagery while simultaneously generating ODs and $D_{ei}$s that show the best match to the INCAS KA PSD observations and ceilometer optical depth retrievals.




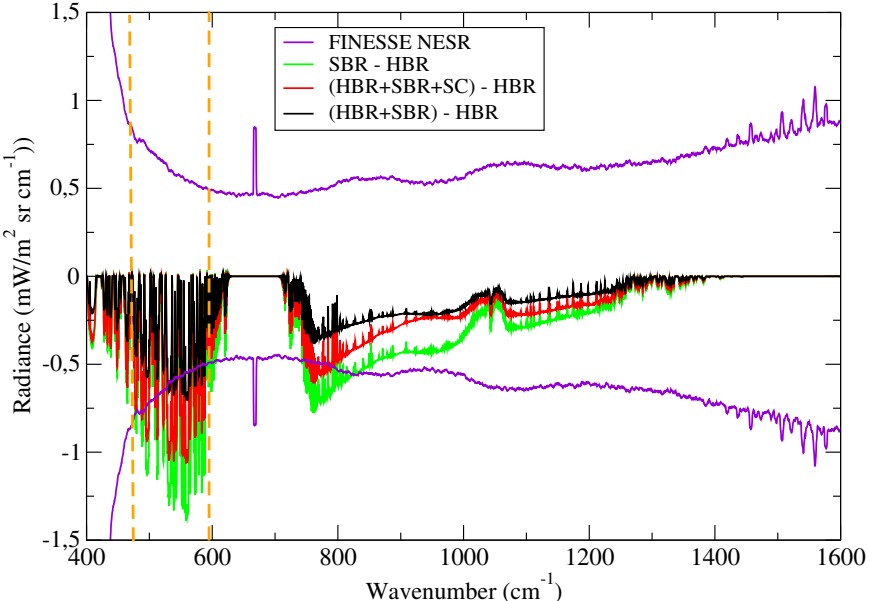

**Figure 16.** Variations of the simulated spectral radiance by varying the habit distributions used in the analysis of FINESSE measurements (SBR (green), HBR+SBR (black), HBR+SBR+SC (red)) with respect to the case of HBR only. Variations are compared with the FINESSE noise (violet). The band of higher sensitivity to crystal shape in the far-infrared is seen between 470-600 cm$^{-1}$ and is delimited by orange dashed lines. From this spectral band we can discriminate the habit crystal composition.

Selecting the HBR habit we further investigate the impact of FIR information on the inferred effective diameters. Fig. 17 repeats the information shown in the top panel of Fig. 15 but superposes the equivalent retrievals obtained if only FINESSE radiances over the spectral range 650-1500 cm$^{-1}$ are considered. It is immediately apparent that removing the FIR information results in a poorer fit to the $D_{ei}$ value derived from the INCAS KA observations, particularly during the latter part of the overpass. Despite the improvement of the retrieval accuracy (the retrieved $D_{ei}$ are more in accordance with the INCAS values), we see a general increasing of the statistical retrieval error when the FIR is used. This can be explained with the increasing of the instrumental noise over this spectral region Rodgers (2000), as visible from Fig. 18.

### 5.1 Discussion

The methodology we chose to validate the scattering properties for ice crystals used in this study is to retrieve all parameters that characterize either the atmosphere or cirrus clouds and then compare the products with the corresponding available measurements. In such a way, we avoid over-constraining the atmospheric state we use to simulate the spectral radiance since the available measurements were not perfectly co-located. For example, in this case the HATPRO was located 1.9 km away while the INCAS KA measurements were not instantaneously taken for the entire cirrus vertical profile that FINESSE was measuring but were rather built up over time following the INCAS KA ascent and horizontal sampling. The FINESSE measurements





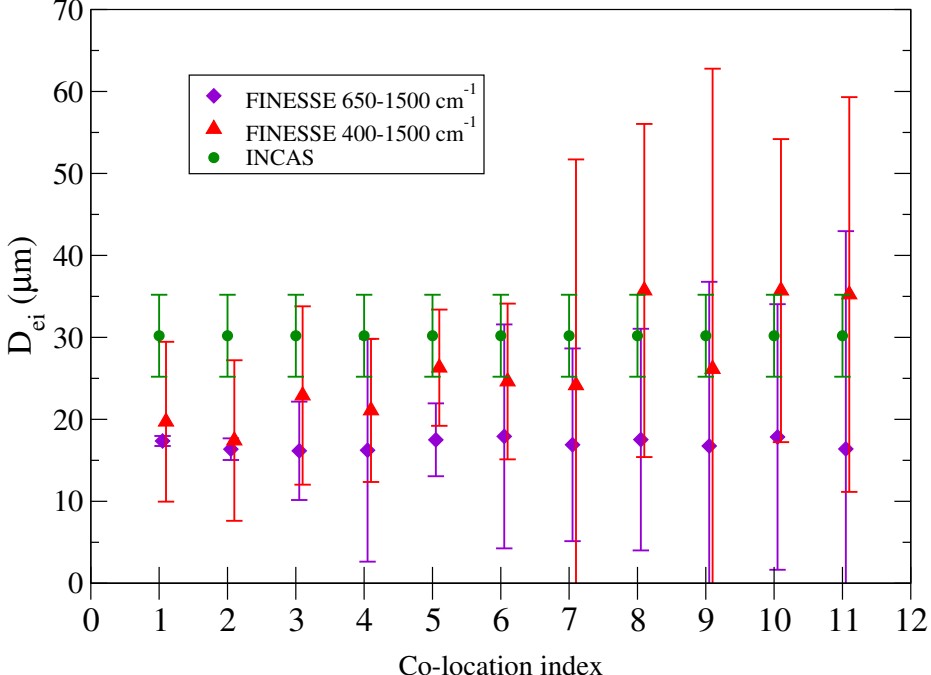

**Figure 17.** Comparison of the ice effective diameters derived from INCAS KA measurements (violet dots) and those retrieved from FINESSE spectra by using the whole spectral band between 400–1500 cm$^{-1}$(red triangles) and cutting the FIR portion below 650 cm$^{-1}$(blue diamonds).

themselves (e.g. Fig. 14) indicate that the cirrus does evolve with time, even over the relatively short period of the INCAS KA observations. To minimise this impact we assume only two parameters from the ancillary measurements: the ice crystal habit inferred from the INCAS KA 2D-S and CPI measurements (hollow bullet rosettes) and the cirrus CTH/CBH derived from ceilometer backscattering profiles. We demonstrate that varying the habit in the simulations and repeating the retrieval procedure results in effective diameters that are not in agreement with the values derived from the PSDs measured by the INCAS

KA.

Given the caveats noted above, the retrievals from FINESSE show excellent consistency with independent realisations of the atmospheric state and measurements of cirrus properties. They also show the critical importance of the FIR to achieving this consistency. Nonetheless, it is useful to consider the fit of the final retrieved spectra to the observations. As an example, Fig. 18 shows the measured FINESSE spectrum, the simulated radiance at the last iteration, and the associated residuals for

scenario 4 (08:25:29 UTC, 30329 sec. since midnight). The retrieved $D_{ei}$ and OD were found equal to (24.2±1.4) $\mu$m and (0.144±0.001), respectively. For all scenarios the final value of the normalized $\chi^2_N$, the cost function in Eq. (13) divided by the number of spectral channels used, ranged between 2.00–2.10.

Fig. 18 shows that the residuals between the measurement and the simulation predominantly fall within the instrument NESR (blue curves). Notable exceptions are seen within the 667 cm$^{-1}$CO$_2$ band wings between 550–600 and 750–780 cm$^{-1}$,





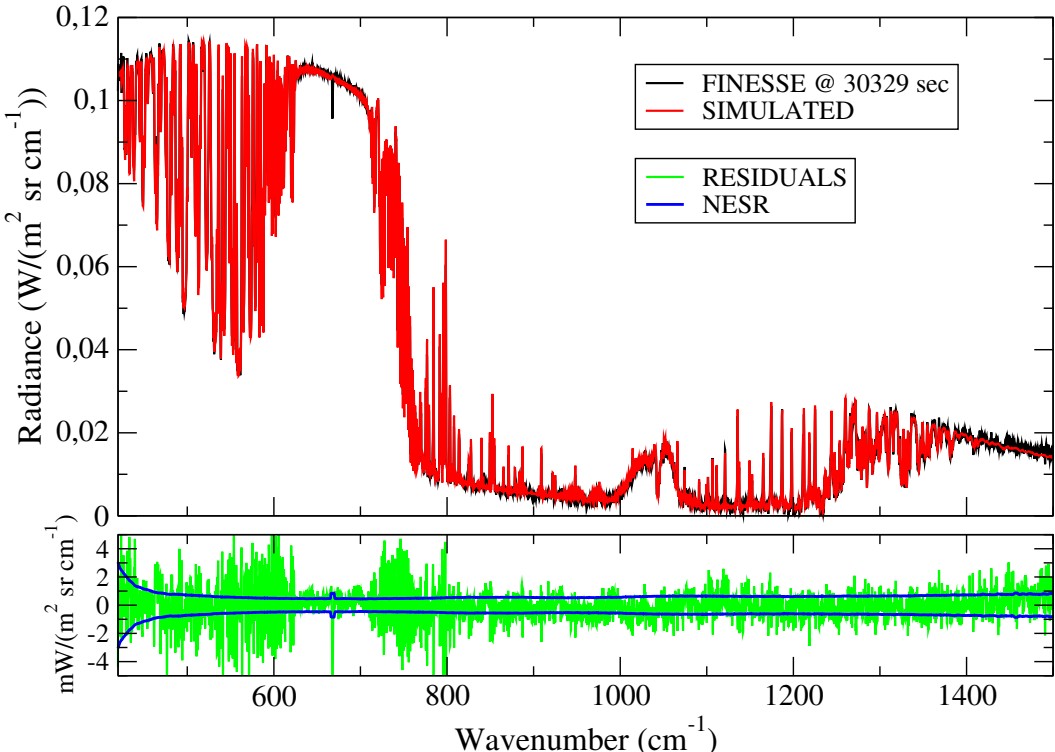

**Figure 18.** Upper panel: comparison of the simulated spectrum at the last retrieval algorithm iteration (red curve) with the FINESSE measured spectral radiance (black curve) at 30329 sec. since midnight (08:25:29 UTC, scenario 4) on 17 February 2023. Lower panel: the residuals (green curve) are reported in comparison with the NESR, that is the statistical instrumental noise.

perhaps because at these wavenumbers, where the spectral lines are very dense, the instrument line shape is not perfectly characterized. This feature appears systematically for each retrieved spectrum, and this is the reason why the $\chi_N^2$ is always slightly larger than 2.0.

    Outside of these regions, in the mid infrared the residuals are very small, indicating that the optical properties are able to correctly reproduce the spectrum. Larger residuals are seen in the FIR region (Fig. 19), mostly due to water vapour lines, but

also within the main micro-windows (centered at 478, 497, 530, 560 cm$^{-1}$, respectively, discussed by Turner (2005)) and shown by the orange vertical lines in Fig. 19. The reduced agreement in the FIR region is consistent with previous studies by Bantges et al. (2020) and Maestri et al. (2014).

## 6   Conclusions

    In this paper we describe the first synergistic retrieval of Arctic cirrus combining information provided by a ground-based

spectrometer, ceilometer and radiometer and in-situ aircraft observations of cloud particle size and shape. Specifically, we





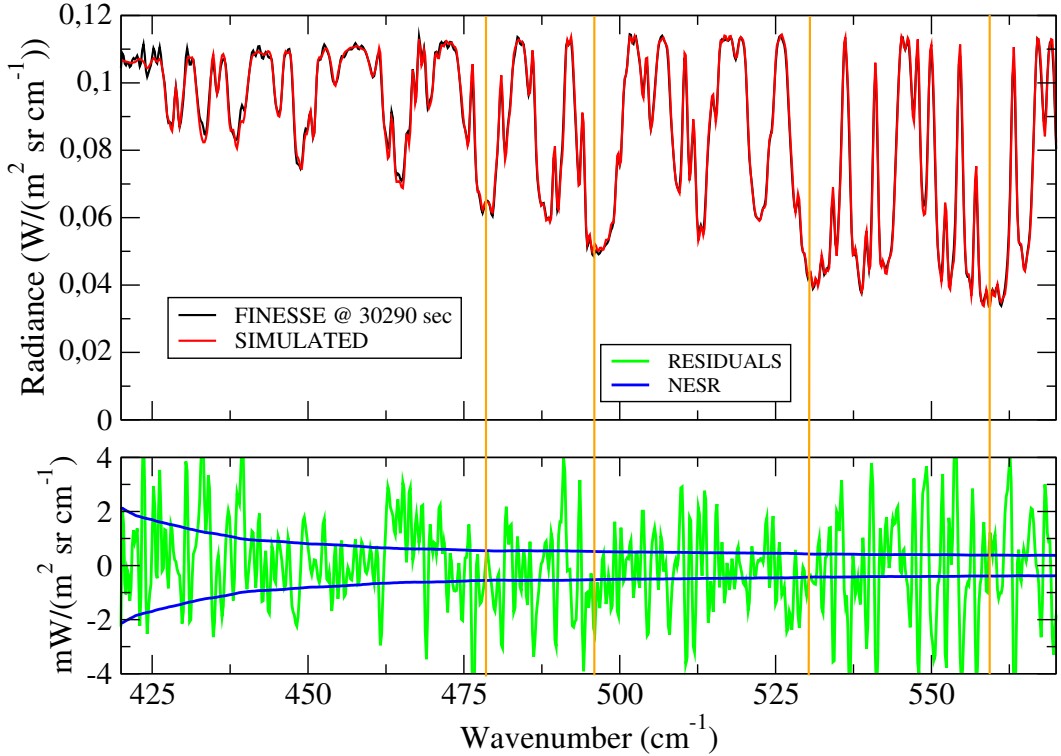

**Figure 19.** Upper panel: close-up in the FIR region of the comparison shown in Fig. 18. Lower panel: the residuals (green curve) are reported in comparison with the NESR and the main FIR microwindows, centered at 478, 497, 530, 560 cm$^{-1}$ (vertical orange lines) as indicated by Turner (2005).

make use of spectrally resolved downwelling radiances measured by the Far-INfrarEd Spectrometer for Surface Emissivity (FINESSE), over the range 400–1500 cm$^{-1}$ in conjunction with cloud vertical structure from a ground-based ceilometer and, uniquely, co-located in-situ measurements from a Hawkeye Combination Cloud Particle Probe and a Cloud, Aerosol and Precipitation Spectrometer mounted on the National Institute for Aerospace Research of Romania (INCAS) aircraft. The

simultaneous measurements span a period of 13 minutes during the morning of February 17th, 2023.

By applying the Simultaneous Atmospheric and Cloud Retrieval (SACR) code, we were able to simulate the FINESSE spectra and retrieve the cirrus cloud effective diameter ($D_{ei}$) and visible optical depth (OD), together with the atmospheric profiles and ozone total column. We selected a subset of 11 spectra between 8:25-8:38 UTC, maximising the overlap with the INCAS aircraft measurements inside the cirrus. Bullet rosettes were identified as a dominant ice crystal habit within the cloud.

The PSDs measured at different heights sampled by the aircraft, were used to derive the average in-situ $D_{ei}$ for comparison with the values obtained from the FINESSE retrievals. Using the optical properties from Yang et al. (2013) and assuming a Hollow Bullet Rosette (HBR) crystal shape, we show the agreement between the retrieved $D_{ei}$ with counterpart derived from



the aircraft observations within measurement and retrieval uncertainty. Varying the habit distribution, for example, by adding different shapes like hexagonal columns, and repeating the retrieval, degraded the agreement.

Observations from the ceilometer allowed us not only to set the cloud top and bottom height in our simulations, but also to derive the OD, by applying the Klett algorithm in comparison with the FINESSE retrieval. We found a good agreement between the two retrieval products both when the cirrus was optically thicker, during the first six FINESSE measurements (mean OD = (0.150±0.005)), and thinner during the last five FINESSE measurements (mean OD = (0.077±0.001)).

    To complete the analysis, we compared the retrieved water vapour and temperature profiles and ozone total column with

co-located values from the ECMWF ReAnalysis-5 (ERA5) database, retrievals from a Humidity and Temperature Profiler (HATPRO) situated at Andøya Space. These comparisons were supplemented, where appropriate, by surface observations from Vaisala probes attached to FINESSE, and by in-cloud temperature measurements from the INCAS aircraft. Both FINESSE retrieved temperatures and ERA5 temperatures showed excellent agreement with those measured by the INCAS probes inside the cirrus. At the surface, the level of agreement appeared to show a dependency on the inferred cirrus OD, with FINESSE

showing closer agreement to the single ERA5 value during the first six (higher OD) measurements before aligning more closely with the Vaisala values. In all cases the ERA5 and retrieved values were colder than the Vaisala temperatures by up to 1.5 K. A similar low bias was seen between the water vapour volume mixing ratio at the surface obtained from ERA5 and that measured by the Vaisala probe. However, in this case the agreement between the probe and the FINESSE retrieval was much closer. An ERA5 low bias relative to FINESSE was also manifested in the retrievals of the total precipitable water vapour (PWV) above

the instrument. In this case, equivalent values derived from the HATPRO support the sense that ERA5 shows a significant humidity underestimate, with PWV retrievals which are closer to those derived from FINESSE, albeit still persistently smaller.

    Finally, to understand the importance of information from the far-infrared in determining the cirrus properties we (a) assessed the radiometric sensitivity to the choice of crystal habit and (b) repeated our HBR retrieval but only considered the spectral range 650–1500 cm$^{-1}$. In common with earlier findings (e.g. Bantges et al. (2020)) the strongest sensitivity to altering the

crystal habit is seen in the 500–600 cm$^{-1}$ region. Moreover, even with the HBR habit selected, when the spectral range is cut to exclude the far-infrared the level of agreement with the aircraft size measurements is significantly reduced. The combination of these two factors suggests the inclusion of the far-infrared is crucial to achieving the improved agreement with the aircraft derived $D_{ei}$ seen for the HBR habit relative to the other habits investigated here. Our results provide further motivation for increasing the number of measurements exploring this portion of the spectrum, in particular from space, as the future ESA

Earth Explorer-9 FORUM mission aims to do.

*Author contributions.* TEXT

    Conceptualization, GDN; Data curation, GDN, JM, ROD, TC, AF, MF, RL, PY; Funding acquisition, HB, HO, DS, LP; Investigation, HB, JM, LW, SP, SM, ROD, TC, SNV, AV, SG, RB, LP; Methodology, GDN; Project administration, HB; Software, GDN ; original draft, GDN; Writing – review & editing, HB, JM, HO. All authors have reviewed and agreed to the

published version of the manuscript.



*Competing interests.* The authors declare that they have no conflict of interest.

*Acknowledgements.* We would like to acknowledge the European Research Council (StG 758005 (Mixed-Phase Clouds and Climate) and CoG 101045273 (STEP-CHANGE)), EEA Grants/Norway Grants (grant no. EEARO-NO-2019-0423/IceSafari, contract no. 31/2020) and EU-HORIZON-WIDERA-2021 (Grant 101079385 (BRACE-MY)) for supporting the campaign and airborne data analysis. We are also
grateful to the INCAS pilots, the staff at Andøya Space and the Norwegian Meteorological Institute for their support during the campaign. Funding for the deployment of FINESSE was provided by the European Space Agency (ESA) under contract no. 4000137153/22/NL/IA. HB, JM and RB were supported by the UK Natural Environment Research Council grant no. NE/Y006216/1. Part of the research activities described in this paper were carried out with contribution of the Next Generation EU funds within the National Recovery and Resilience Plan (PNRR), Mission 4 - Education and Research, Component 2 - From Research to Business (M4C2), Investment Line 3.1 - Strengthening and
creation of Research Infrastructures, Project IR0000038 – "Earth Moon Mars (EMM)". EMM is led by INAF in partnership with ASI and CNR. Finally, we also thank the Norwegian Meteorological Institute for making the Ceilometer data available.



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
