# Peer review of "Achieving consistency between in-situ and remotely sensed optical and microphysical properties of Arctic cirrus: the impact of far-infrared radiances"

_EGUsphere, 2025_

## Author Comment (AC1)

**Answers to reviewer 2**

**October 31, 2025**

This paper addresses a critical issue for passive remote sensing of clouds from infrared spectral measurements. Previous measurements (e.g., Bantges et al 2020) indicated inconsistency in the ice droplet properties in the current database, which leads to difficulty in achieving radiance closure across the visible-MIR-FIR spectra in the measurements. This paper revisits this inconsistency issue using the measurements from a new campaign with a different deployment strategy, i.e., measuring cloud spectra from ground as opposed to airborne. I found this study has a well-motivated objective and a great potential to help found methods utilizing future FIR satellites such as FORUM for cloud remote sensing. However, I also found the current manuscript has several major issues. The presentation and interpretation of some results especially raised concerns, which should be addressed before it is published.

We sincerely thank the reviewer for the careful, thorough, and constructive evaluation of our manuscript. The review highlighted important points of uncertainty that required clearer explanation, allowing us to improve the clarity and rigor of our work. At the same time, we greatly appreciate the reviewer's recognition of the potential of this study and the value it can offer to the scientific community. The insightful comments have been instrumental in strengthening and improving the manuscript.

A major comment is that this work, as it currently stands, shouldn't be mistaken as a retrieval test that proves or disapproves the feasibility of retrieving cloud microphysical properties from the FIR spectral measurements. In essence, this is a radiance closure test, showing sensitivity to ice habit as well as other potentially retrievable parameters. If the authors intend to establish this as a retrieval work, the paper needs to more systematically and comprehensively test whether and how the cloud microphysical properties (size, habit, etc) can be "unambiguously" determined from macrophysical ones (optical depth, temperature, etc) and from other atmospheric states (temperature, humidity, ozone, etc). This should be done using both synthetical data with perfectly known truth to quantify such quantities as averaging kernels and degrees of freedom for signals (DFS) and using independently measured cloud data to validate retrievals from actual spectral measurements. For example, the use of atmospheric profiles from ERA5 as both initial guess and validation is unfit to represent an actual retrieval. Moreover, although varying the multiple parameters is shown to lead to reasonable agreement between simulated and measured spectra, there is no proof that there is enough information content to determine them altogether in a real retrieval. In fact, some results, such as the noticeable differences in the sizes "retrieved" when different habits are assumed (Fig 15), suggest there is substantial degeneracy. If this (retrieval) isn't the intention, the paper should be clear about it and avoid misleading claims such as "first retrieval" in Abstract (Line 1) and throughout the paper. It would be more appropriate to state the tests as "adjusting", as opposed to "retrieving", the relevant cloud parameters.

The reviewer is correct in stating that this is a kind of radiative closure test, which is performed using the retrieval technique, but only to "adjust" the state vector of atmospheric parameters among the local minima of the  $\chi^2$ , because of local variability that cannot be perfectly reproduced. The inversion algorithm allows us to overcome this issue while maintaining consistency between in situ/remote sensing measurements and the model. As stated in our response to Reviewer 1, the retrieval of atmospheric profiles is crucial to achieving consistency among all the available measurements. The reviewer is correct in noting that we lack sensitivity to cloud altitude; however, that is not the goal of our analysis. The reason for using the inversion technique is to "adjust", as correctly suggested by the reviewer, the parameters around the minimum of  $\chi^2$ , since we cannot know, within the associated uncertainties, how much the measurements should vary in order to achieve full consistency. The retrieval is the only correct and rigorous procedure to do that. This is because local factors and variables come into play that are difficult to account for when setting up the simulation framework, even when attempting to incorporate all available information. We

have made revisions to the text, emphasizing that the retrieval is not the main objective of the work, but rather a means to achieve consistency between the measurements and the model.

We changed the reference to retrieval in the abstract but also in the rest of the manuscript. In particular in the Introduction we emphasized that the sensitivity of FIR to distinguish the habit composition has already shown in previous studies with synthetic observations and a previous consistency study between cloud properties was already performed from Antarctica by using ground-based instruments but, critically, not in situ observations. The sentence we added in the Introduction is the following: "We use these data to provide a well-characterised test of our ability to achieve consistency between measured and retrieved atmospheric temperature and water vapour profiles, ozone total columns and cloud optical and micro-physical properties given currently available parameterizations of ice crystals, with a particular emphasis on demonstrating observationally how FIR radiances can constrain crystal habit. This aspect is well known and has been studied in the literature (Di Natale et al., 2024). However, to the best of our knowledge, the capability to achieve a better consistency with one habit distribution compared to another has never been demonstrated. Furthermore, a previous study of the consistency between the optical and microphysical properties of clouds, exploiting spectral radiance measurements in the FIR together with ground-based data such as radar reflectivity and particle imaging, has already been carried out from Antarctica (Di Natale et al., 2022). However, in these studies remotely sensed cloud microphysics (particle size and habit) were only compared with ground-collected measurements of precipitating ice crystals: they have never before been compared with in situ data acquired inside cirrus clouds, as is done in the present work." Also in Conclusions we have changed the initial sentence as follows: "In this paper, we present the first demonstration of consistency between atmospheric measurements and Arctic cirrus cloud parameterizations by combining information from a ground-based spectrometer and sensors, ceilometer, and radiometer with in situ aircraft observations of cloud particle size and habit.".

Tests demonstrating the robustness of the SACR inversion algorithm across various atmospheric and cloud scenarios have already been conducted and published in several previous works, both using synthetic simulations (Di Natale et al., 2020) and real measurements from different field campaigns Di Natale et al. (2021); Palchetti et al. (2016). In particular, extensive analyses have been carried out in Antarctica, where 11 years of spectral radiance data from the REFIR-PAD spectrometer have been examined. In Di Natale et al. (2020) the first 4 years are published, the complete dataset of 11 years is coming up for publication. This instrument also operates in the FIR, extending down to 100 cm-1. Moreover, a retrieval algorithm specifically designed to derive the habit distribution has already been implemented and validated using synthetic simulations in nadir-viewing configuration for the FORUM mission observations (Di Natale et al., 2024). The results demonstrated that the FIR radiance, when associated with noise of this magnitude, that is similar to FINESSE's above 450 cm-1, exhibits sufficient sensitivity to the ice crystal habits.

As noted in our response to reviewer 1, we have introduced a sensitivity study based on the vertical distribution of the averaging kernels (AK) for temperature and water vapor, as shown in Fig. 1, where it can be seen that the sensitivity for temperature and water vapor is mainly limited to about 1 km and 4 km above the surface, respectively. So, as the reviewer pointed out, there is not much sensitivity to the internal temperature of the cloud. The average degrees of freedom for signal (i.e., the trace of the AK) were found to be 3 and 7, respectively. For this reason, the vertical levels were limited to about 1 km for temperature (3 levels) and to about 9 levels for water vapor below 4 km.

This choice allows us to adjust the profiles around a minimum that is assumed to be close to, but not exactly coincident with, the ERA5 profiles — as indeed turns out to be the case. In this context, there is strong motivation for this approach: ERA5 profiles are not radiosonde measurements but are derived from models and data assimilation. Nevertheless, even radiosondes may not be exactly co-located with the measurements and can be affected by small biases and local variability in the lower atmospheric layers.

For these reasons, we believe that presenting the retrieval of the atmospheric profiles is an integral part of the study and can help colleagues when considering how to approach similar studies in future, so should not be overlooked.

We have added a discussion on this aspect and included the corresponding figure in section 3.1.3 of the manuscript. ERA5 profiles are not assumed as a priori/initiale guess, to generate these we have perturbed the ERA5 baseline by values reported in Table 2.

The reviewer's points regarding information content (IC) is correct; therefore, we show the distribution of the IC, the correlations between  $D_{ei}$ -OD, and the number of degrees of freedom for the different habits. We find that there are essentially no differences in IC and DOFs — up to 0.3/0.4 and  $10^{-4}$ , respectively — and the differences in correlations are smaller than 0.25. The plots are shown in Fig. 2. This indicates that, a priori, there is no greater or lower sensitivity in distinguishing cloud parameters when using different habit distributions. Please see our responses to reviewer 1 for further comments around this general point.

Another major comment is that the paper doesn't provide a clear answer to the motivating questions. For example, is the pan-spectral inconsistency pointed out in earlier works (e.g., Bantges et al) now reconciled, e.g., by using the ground measurements or by considering a HBR habit? Given that

Figure 1: Left panel: vertical distribution of averaging kernels of temperature. Right panel: same for water vapour.

some coauthors here were those who championed the earlier works, I would very much like to see such answers, which would help put this work in context and better identify its value. To these questions, combining Fig 16 with Fig 18/19 and showing how the residuals can (or cannot) be minimized by using different residuals would be helpful. Regarding the ground-based measurement strategy, I would appreciate more discussions on its advantage and disadvantages compared to the airborne approach. For example, the cirrus as visualized in Fig 2 indicates clearly spatial inhomogeneity and, as noted in the paper, varies in measurement time; how does this affect sampling consistency and representativeness of retrieval (e.g., the optical depth) from spectra collected over a finite (200-m) FOV? Reflections on these questions and/or suggestions for future campaigns would be especially useful. A naïve question is: did the airplane affect the cloud fields and exacerbate the inhomogeneity issue, as indicated by the contrails? Fig 16: There seems to be equally discernible signals in MIR (around 800 cm-1), which raises the question on the FIR benefits claimed.

As we also pointed out to Reviewer 1, a dataset of 11 measurements cannot, of course, be representative of the wide variety of scenarios that can occur in nature. However, the range of optical depths between 0.05 and 0.2 represents a typical thin-cirrus case and constitutes a first attempt to demonstrate that it is possible to achieve consistency in the parameterization of cloud microphysics with in situ measurements, although we see that some inconsistencies in the modeling of FIR radiances are still present.

It should also be emphasized that simultaneous measurements of this kind are still rare to date, as they are clearly expensive and logistically challenging. Therefore, we believe that every small step in this direction provides valuable progress in the understanding and modeling of clouds, even with the awareness that much remains to be done. We have attempted to stress this in the revised manuscript.

Regarding the issue of inhomogeneity raised by the reviewer, we can respond that this is a very complex aspect to address and has been extensively studied, starting from satellite observations, where it is necessary to estimate the level of pixel contamination in the signal measured by the instrument (Sgheri et al., 2022). However, unlike those cases — where the area observed by the instrument at cloud level can span several kilometers — in ground-based observations this effect is significantly reduced. In fact, for FINESSE, as mentioned earlier, the field of view (FOV) at cloud altitude is less than 200 m (approximately 180 m), which corresponds to about 1/40 of the circular area defined by the aircraft track shown in Fig. 2. Therefore, the effect of inhomogeneity is strongly mitigated, and we can reasonably assume that the FOV coverage is homogeneous to a good approximation in this case.

To better clarify the imperfect consistency between the simulations and the spectral measurements in the FIR, we calculated, in agreement with previous studies, the mean residuals of all simulations with respect to their corresponding measurements. These were then compared with the mean squared error - defined in Eq. 20 in the text - derived from NESR and calibration, in 13 microwindows used in previous studies (Turner, 2005) as shown in Fig. 4: the root mean square (RMS) of the average residuals in radiance are compared with the average noise in each micro-window. The comparison is shown in Fig. 3, while Fig. 4 presents a histogram of the mean (root mean square, RMS) residual values relative to the measurement error, clearly showing that, while the values fall fully within the error in the MIR, they do not in the FIR. Fig. 4, along with the corresponding explanation, has been added to the text, replacing Fig. 19.

Fig 17 doesn't show convincingly different performance between the two results (with vs without FIR). Even if it did, this wouldn't make strong evidence given the aforementioned degeneracy and strong sensitivity to habit. To elucidate this point, a formal retrieval assessment based on DFS as suggested above would be more quantitative and convincing.

As also replied to reviewer 1, in contrast to the case with different habit distributions, when we consider whether or not to include the FIR component in the retrieval procedure, Fig. 5 shows that IC changes significantly, decreasing from 9 to 6; the correlation between  $D_{ei}$  and OD parameters increases notably to about 0.8/0.9, and the difference in DOFs in this case is on the order of  $10^{-2}$  — that is, one hundred times larger than in the case of different habits.

We performed several retrieval tests simulating FINESSE observations, both for the values obtained from the analysis in this work and for other cases with larger optical depths and particle effective diameters. Specifically, we tested the capability of the retrieval algorithm to recover the "true" values with and without including the FIR spectral region. To this end, we generated synthetic observations using the FINESSE noise and two different atmospheric profiles, together with varying cloud parameters — optical depths ranging from 0.07 and 0.18 (representative of the values obtained from the FINESSE measurements) up to 0.8 and 1.5 — while fixing the  $D_{ei}$  value at 30  $\mu m$ .

Our tests demonstrated that the retrieval algorithm is capable of reproducing the true state from synthetic observations whether or not we include the FIR if we fit the observations with habit distribution equal to those used to generate the "truth", as shown in Fig. 6 and 7. However, when we invert the real measurement, we find that we can only match the values measured by INCAS when the FIR is included. This is likely due to the fact that the actual microphysics of the cloud is more complex than what our models can reproduce, and by adding spectral information in the FIR we can approach more closely the absolute minimum of the  $\chi^2$  distribution.

The fact that retrievals performed while neglecting the FIR portion yield underestimated effective diameter values compared to those obtained when including the FIR is likely due to the actual microphysical structure of the cloud being more complex than what can be represented by our parameterization. However, a much more detailed investigation would be required to fully explore this, which is beyond the scope of the present work, although it remains an interesting topic for future research. We added the following sentence in the text in section 5: "We would like to emphasize that, based on the FINESSE measurements presented and analyzed, performing the inversion with limited information that excludes the FIR leads to an underestimation of the effective diameter values, but this result cannot, at this time, be generalized to every possible scenario. This clearly calls for a more in-depth investigation, which will be addressed in future work".

In addition, we have modified the abstract as follows: "Furthermore, we show that the radiance information contained within the far-infrared (wavenumbers; 650 cm-1) spectrum is critical to achieving this level of agreement with the in-situ aircraft observations. The results emphasize why it is vital to expand the current limited database of measurements encompassing the far-infrared spectrum, particularly in the presence of cirrus, to explore whether this finding holds over a wider range of conditions."

**References**

- Di Natale, G., Bianchini, G., Del Guasta, M., Ridolfi, M., Maestri, T., Cossich, W., Magurno, D., and Palchetti, L.: Characterization of the Far Infrared Properties and Radiative Forcing of Antarctic Ice and Water Clouds Exploiting the Spectrometer-LiDAR Synergy, Remote Sensing, 12, https://doi.org/10.3390/rs12213574, 2020.
- Di Natale, G., Palchetti, L., Bianchini, G., and Ridolfi, M.: The two-stream  $\delta$ -Eddington approximation to simulate the far infrared Earth spectrum for the simultaneous atmospheric and cloud retrieval, Journal of Quantitative Spectroscopy and Radiative Transfer, 246, 106 927, https://doi.org/https://doi.org/10.1016/j.jqsrt.2020.106927, 2020.
- Di Natale, G., Barucci, M., Belotti, C., Bianchini, G., D'Amato, F., Del Bianco, S., Gai, M., Montori, A., Sussmann, R., Viciani, S., Vogelmann, H., and Palchetti, L.: Comparison of mid-latitude single- and mixed-phase cloud

- optical depth from co-located infrared spectrometer and backscatter lidar measurements, Atmospheric Measurement Techniques, 14, 6749–6758, https://doi.org/10.5194/amt-14-6749-2021, 2021.
- Di Natale, G., Turner, D. D., Bianchini, G., Del Guasta, M., Palchetti, L., Bracci, A., Baldini, L., Maestri, T., Cossich, W., Martinazzo, M., and Facheris, L.: Consistency test of precipitating ice cloud retrieval properties obtained from the observations of different instruments operating at Dome C (Antarctica), Atmospheric Measurement Techniques, 15, 7235–7258, https://doi.org/10.5194/amt-15-7235-2022, 2022.
- Di Natale, G., Ridolfi, M., and Palchetti, L.: A new approach to crystal habit retrieval from far-infrared spectral radiance measurements, Atmospheric Measurement Techniques, 17, 3171–3186, https://doi.org/10.5194/amt-17-3171-2024, 2024.
- Palchetti, L., Natale, G. D., and Bianchini, G.: Remote sensing of cirrus microphysical properties using spectral measurements over the full range of their thermal emission, J. Geophys. Res., 121, 1–16, https://doi.org/10.1002/2016JD025162, 2016.
- Sgheri, L., Belotti, C., Ben-Yami, M., Bianchini, G., Carnicero Dominguez, B., Cortesi, U., Cossich, W., Del Bianco, S., Di Natale, G., Guardabrazo, T., Lajas, D., Maestri, T., Magurno, D., Oetjen, H., Raspollini, P., and Sgattoni, C.: The FORUM end-to-end simulator project: architecture and results, Atmospheric Measurement Techniques, 15, 573–604, https://doi.org/10.5194/amt-15-573-2022, 2022.
- Turner, D. D.: Arctic mixed-Phase cloud properties from AERI lidar observation: algorithm and results from SHEBA, Journal of Applied Meteorology, 44, 427–444, 2005.

Figure 2: Upper panel:Information content of retrieval for the 4 habit distributions, corresponding to the different colors, considered in the analysis. Middle panel: degrees of freedom. Bottom pane: correlations  $OD-D_{ei}$ .

Figure 3: Average residuals (black circles) are reported in comparison with the average noise  $\sigma_{av}$  (red curve) in 13 main microwindows discussed by Turner (2005) indicated by the center band (blue dashed line).

Figure 4: Histogram of the comparison between the root mean square (RMS) of the residuals (blue boxes) and the average noise over all scenarios (orange boxes) in the 13 selected microwindows.

Figure 5: Upper panel:Information content of retrieval for the case FIR was considered (blue) and neglected (orange). Middle panel: degrees of freedom. Bottom pane: correlations  $OD-D_{ei}$ .

Figure 6: Top panel: comparison of the retrieved  $D_{ei}$  by including the FIR (blue dots) wit respect to the "truth" (red curve). Bottom panel: same but neglecting the FIR (green dots). In both cases the initial guess and a apriori is 80  $\mu$ m as done with the analysis.

**Scenario: Dei30\_OD0.18**

**Scenario: Dei30\_OD0.18\_650-1500cm-1**

Figure 7: Top panel: fit and residuals by using the FIR for the case OD = 0.18 and  $D_{ei}=30~\mu\mathrm{m}$  . Bottom panel: same by neglecting the FIR.

---

## Author Comment (AC2)

**Answers to reviewer 1**

**October 31, 2025**

This paper describes an intercomparison of the properties of cirrus measured by in-situ probes vs those retrieved from a ground-based infrared spectrometer that spans from the mid-infrared to the far-infrared (i.e., from 1500 cm-1 to 400 cm-1). The authors use a ceilometer to ascertain the base and top of the cirrus clouds, and to estimate their optical depth. Lastly, their infrared retrieval framework also retrieves thermodynamic profiles, which they compare against ERA5 model output and retrieved profiles from a collocated ground-based microwave radiometer. The paper achieves what it sent out to do, namely, to perform an intercomparison of derived cirrus properties, which includes an evaluation of the assumption of the habit of the ice particles, and thermodynamic profiles. It is well written, and the references included are sufficient.

We sincerely thank the reviewer for providing a precise and insightful summary of our manuscript. The careful evaluation not only highlighted aspects requiring clearer explanation but also recognized the validity and potential impact of our work. We greatly appreciate the thoughtful comments, which have helped us improve the clarity and strength of the manuscript.

My main concern is that this intercomparison evaluate 11 samples that were collected within a 30-minute period on a single day at a single location. Thus, I must ask: how representative are these results? Would these same results hold for different atmospheric state conditions (i.e., temperature and humidity profiles) or cirrus conditions (i.e., optically thicker clouds, cirrus generated by other means, etc). This is further complicated by the sky image shown in Figure 2, which suggests that their in-situ aircraft is leaving a contrail (based upon the flight pattern on the right and the clear spiral signature in the sky image). Can we assume the ice habit in the contrail is consistent with the habit of the background cirrus? How much is the contrail impacting either the radiometric observations made by the FINESSE or the in-situ obs (when aircraft starts its next circle)? Additionally, the ice optical depths sampled here are small; less than 0.2 – so again, do the conclusions (e.g., the importance of the far-infrared channels) hold when in cases when the optical depth is larger? Before this paper should be accepted, I would hope that the authors can address these concerns well.

We understand the reviewer's concern and we agree that measurements limited to a time interval of less than one hour are not representative of a wide range of atmospheric and cloud scenarios. However, the purpose of this work is to assess as much as possible the consistency of spectral radiance simulations using currently available databases of the optical properties of ice clouds, in combination with both in situ and remote sensing measurements of high-altitude ice clouds. Unfortunately, the great difficulty of performing such measurements — especially in situ observations with aircraft — means that they are scarce and always time-limited. The availability of in situ aircraft measurements of the microphysics of ice particle size distributions, together with an estimate of crystal habits within the cloud, represents a unique asset that must be exploited as much as possible, especially when simultaneous remote sensing measurements across different spectral bands are available, even though the amount of data is not as large as desired.

In addition, the capability to correctly retrieve cirrus properties under a wide range of both atmospheric conditions (temperature and water vapor) and cloud conditions (particle size and optical depth) has already been clearly demonstrated through results that are partly published and partly in preparation, based on the analysis of measurements from the REFIR-PAD spectrometer (a Fourier spectrometer similar to FINESSE covering the FIR spectrum) at Dome-C (Antarctica) in the presence of a backscattering and depolarization lidar. Within such studies, test of consistency have already performed by using various ground-based measurements including also radar, covering different cloud scenarios, but never available unique and precious in situ measurements from aircraft or balloon sounding inside the clouds. In previously published studies on the analysis of REFIR-PAD data, (e.g. Di Natale et al. (2020); Maestri et al. (2014)), a non-perfect matching of the residuals in the FIR has already been observed — although it is less evident due to the higher instrumental noise of REFIR-PAD compared to FINESSE in the spectral range above 400

 ${\rm cm}^{-1}$ . This same result is also found in the present work, even though the residuals are averaged over a much more limited set of measurements than those collected at Dome-C.

From REFIR-PAD measurements, we were able to produce a Level-2 dataset of atmospheric profiles and properties of ice and mixed-phase clouds over 11 years of observations, from 2012 to 2022, covering all seasons with a measurement frequency of 12 minutes. The retrieval products have been thoroughly validated using local radiosonde data, and consistency in the microphysical properties of precipitating clouds has been confirmed through measurements from an ICE-CAMERA — capable of collecting precipitating ice crystals — and a K-band radar, both also installed at Dome C.

We state that we aim at achieving consistency for cirrus clouds. Cirrus are generally optically thin clouds, ranging from subvisible up to 0.3-0.5 in optical depth; for larger values we are already talking about thick ice clouds. Our analysis shows the occurrence of two clearly distinguishable cloud scenarios in which the optical depth of ice clouds decreases, in fact halving from 0.17 to about 0.07/0.08. Thus, the cloud evolves from thin to very thin, covering two rather different values. While this is certainly not representative of a broader range of cases, it nevertheless spans a range of optical depths that well characterize this type of cloud, which permanently envelops the planet.

The scarcity of this type of measurement, due also to their complexity and operational difficulty, highlights the need to fund new campaigns aimed at confirming the results obtained and, if possible, expanding the range of case studies.

Regarding the reviewer's concern about the impact of the aircraft contrail on the retrieval of the measurements, we tried to trace the flight path from the beginning of the acquisition, as shown in Fig. 1 (which has been added to the manuscript), and verified that the contrail did, in fact, intersect FINESSE's field of view at 08:25 UTC — when the thicker cirrus clouds were passing — while in the second measurement sequence, where the cirrus was thinner, there was no overlap.

For this reason, we first decided to test the impact on the simulated FINESSE radiance when introducing a droxtal component for small crystal lengths below 50  $\mu$ m. We verified that a resonance occurs for OD = 0.2 and Dei = 20  $\mu$ m, where the differences between the radiances exceed the instrumental noise, as shown in Fig. 3.

So we regenerated the database of mean optical properties by adding, for smaller particles (below 50  $\mu$ m) a droxtal component to account for the fact that aircraft engines produce small droplets or microparticles that can act as condensation nuclei, promoting the formation of embryonic ice crystals. The most plausible particle shape in this case is therefore that of a hexagonal-faced spheroid, i.e., a droxtal. The new habit distribution is then represented by:

$$f_h(L) = \begin{cases} BR & L > 50 \,\mu\text{m} \\ \frac{1}{2}DX + \frac{1}{2}PL & @ 8:25 \,\text{UTC}, \quad PL & @ 8:37 \,\text{UTC} & L \le 50 \,\mu\text{m} \end{cases}$$
(1)

and similarly for the other  $f_{2,3,4}$  as reported in the manuscript.

We then repeated the retrieval. The results obtained by assuming the two habit composition for smaller crystal lengths are shown in Fig. 2. For the first sequence when the INCAS contrails intersected FINESSE's FOV we see that there is not much effect of introducing the droxtal component, however in the second sequence, when there was no contrail overlap, we obtain much better agreement with the INCAS retrieved  $D_{ei}$  with the dtroxtal component included. We updated Figs. 17 and 19 in the manuscript showing only the retrievals with droxtal+plate for smaller lengths for the first sequence and only plates for the second one. We also added the calculated standard deviation ( $\simeq 10~\mu m$ ) of the INCAS  $D_{ei}$  in Fig. 2 along with the corresponding error, showing that all retrieved diameters in when assuming HBRs are consistent with the internal variability of the INCAS data. As a consequence of the new retrieval we also updated Figs. 14 and 15 of the manuscript. For convenience and simplify, and to make the procedure more streamlined and faster, we use the sum of square root of nesr and calibration error in the VCM of measurements. This has been explained in the manuscript in section 3.1.3.

**Minor concerns:**

Fig 15 (question 1): why do the INCAS points change mean values and have different error bar ranges in the 4 panels? This is most easily seen looking at the bottom-most panel against any of the other three?

The mean value must change because it is calculated using different habits or mixtures of habits. Different habits have different geometric properties. As a result, for a given maximum crystal length, the corresponding values of volume and projected area differ. Consequently, the effective diameter, calculated as reported in Eq. (1), changes.

Regarding the associated error, the reviewer is correct, and we thank them for pointing this out, as it was indeed a mistake. The error associated with the mean was uniquely estimated in Section 3.4, and therefore it does not change

with the variation of the calculated diameter value. The error only affects the first value in the second panel (case  $\frac{1}{2}$ HBR+  $\frac{1}{2}$ SBR). We have corrected this accordingly.

Fig 15 (question 2): Why are the FINESSE error bars so much larger for cases 8-11 when HBR is assumed relative to when you assume  $\frac{1}{2}$ HBR+  $\frac{1}{2}$ SBR? This does not make sense to me.

This is not intuitive, but in the simulations  $\frac{1}{2}HBR + \frac{1}{2}SBR$  the error bars are larger due to the calibration error. The error on the diameter is strongly correlated with the error on the slope of the spectrum in the atmospheric window. This is mainly controlled by the calibration error which has a more pronounced slope than the NESR in this region. For lower OD values this manifests as an enhanced error in the retrievals.

You are using a Bayesian retrieval framework, which is able to provide degrees of freedom for signal. For these very small optical depths, what is the DFS for the retrieved effective radius? Does it change when you assume different habits? Does it change when you only use the midinfrared vs when you use both the midinfrared and far infrared? (This would be a pretty convincing point to make, if the far infrared is indeed important).

The reviewer's point is correct; therefore, we show the distribution of the information content (IC), the correlations between  $D_{ei}$ —OD, and the number of degrees of freedom for the different habits. We find that there are essentially no differences in IC and DOFs — up to 0.3/0.4 and  $10^{-4}$ , respectively — and the differences in correlations are smaller than 0.25. The plots are shown in Fig. 4. This indicates that, a priori, there is no greater or lower sensitivity in distinguishing cloud parameters when using different habit distributions.

In contrast, when we consider whether or not to include the FIR component in the retrieval procedure, Fig. 5 shows that IC changes significantly, decreasing from 9 to 6; the correlation between  $D_{ei}$  and OD parameters increases notably to about 0.8/0.9, and the difference in DOFs in this case is on the order of  $10^{-2}$  — that is, one hundred times larger than in the case of considering different habits.

We performed several retrieval tests simulating FINESSE observations, both for the values obtained from the analysis in this work and for other cases with larger optical depths and particle effective diameters. Specifically, we tested the capability of the retrieval algorithm to recover the "true" values with and without including the FIR spectral region. To this end, we generated synthetic observations using the FINESSE noise and two different atmospheric profiles, together with varying cloud parameters — optical depths ranging from 0.07 and 0.18 (representative of the values obtained from the FINESSE measurements) up to 0.8 and 1.5 — while fixing the  $D_{ei}$  value at 30  $\mu m$ .

Our tests demonstrated that the retrieval algorithm is capable of reproducing the true state from synthetic observations whether or not we include the FIR if we fit the observations with habit distribution equal to those used to generate the "truth", as shown in Fig. 6 and 7. However, when we invert the real measurement, we find that we can only match the values measured by INCAS when the FIR is included. This is likely due to the fact that the actual microphysics of the cloud is more complex than what our models can reproduce, and by adding spectral information in the FIR we can approach more closely the absolute minimum of the  $\chi^2$  distribution.

In conclusion, we can state that using a more comprehensive set of spectral information, including the FIR, allows for more stable solutions that better represent reality. However, a much more detailed investigation would be required to fully explore this, which is beyond the scope of the present work, although it remains an interesting topic for future research. We added the following sentence in the text in section 5: "We would like to emphasize that, based on the FINESSE measurements presented and analyzed, performing the inversion with limited information that excludes the FIR leads to an underestimation of the effective diameter values, but this result cannot, at this time, be generalized to every possible scenario. This clearly calls for a more in-depth investigation, which will be addressed in future work". In addition, we have modified the abstract as follows: "Furthermore, we show that the radiance information contained within the far-infrared (wavenumbers | 650 cm-1) spectrum is critical to achieving this level of agreement with the in-situ aircraft observations. The results emphasize why it is vital to expand the current limited database of measurements encompassing the far-infrared spectrum, particularly in the presence of cirrus, to explore whether this finding holds over a wider range of conditions."

Fig 17: caption errors: INCAS measurements are green dots, and only using the mid-IR is the violet diamonds

We thank the reviewer, we have remade the Fig. 17, now Fig. 19 in the manuscript.

You spend very little time talking about the thermodynamic retrieval and its accuracy. Personally, I do not feel it adds anything to the paper; indeed, it is more distracting. If you elect to keep that

portion in the paper for the next iteration, I would want to see much more discussion about it (recognizing that you don't have strong sources of truth). But, in particular, your statement on line 468 about "excellent agreement...inside the cirrus" needs to be tempered. Does the FINESSE retrieval actually have any information content (i.e., degrees of freedom for signal) at that altitude? I would be very surprised if it did. And if the DFS is very small, then the agreement in temperature within the cirrus is more of a happy coincidence (provided by the statistics of the prior dataset used in the constraint, which you did not discuss at all) then real skill by the retrieval.

The retrieval of atmospheric profiles is crucial to achieving consistency among all the available measurements. The reviewer is correct in noting that we lack sensitivity to cloud altitude; however, that is not the goal of our analysis. The reason for using the inversion technique is to "adjust", as also suggested by reviewer 2, the parameters around the minimum, since we cannot know, within the associated uncertainties, how much the measurements should vary in order to achieve full consistency among them. This is because local factors and variables come into play that are difficult to account for when setting up the simulation framework, even when attempting to incorporate all available information.

For this reason, when we say "very good agreement", we mean that there is consistency between the temperature measurements inside the cirrus and the scaled temperature profile above the last fitted level. We understand, however, that this phrase may be misunderstood, and we have therefore sought to soften and clarify it. We have rephrased it as follows in the text: "The retrieved temperature profile above the last fitted level is merely scaled but it turned out consistent with the temperatures measured by INCAS probe inside the cirrus, and the vertical profile provided by ERA5."

The a priori/initial guess of the atmospheric profiles are now reported in Table 2.

We have introduced a sensitivity study based on the vertical distribution of the averaging kernels (AK) for temperature and water vapor, as shown in Fig. 8, where it can be seen that the sensitivity for temperature and water vapor is mainly limited to about 1 km and 4 km above the surface, respectively. So, as the reviewer pointed out, there is not much sensitivity to the internal temperature of the cloud. The average degrees of freedom for signal (i.e., the trace of the AK) were found to be 3 and 7, respectively. For this reason, the vertical levels were limited to about 1 km for temperature (3 levels) and to about 9 levels for water vapor below 4 km as stated in the manuscript.

This choice allows us to adjust the profiles around a minimum that is assumed to be close to, but not exactly coincident with, the ERA5 profiles — as indeed turns out to be the case. In this context, there is strong motivation for this approach: ERA5 profiles are not radiosonde measurements but are derived from models and data assimilation. Nevertheless, even radiosondes may not be exactly co-located with the measurements and can be affected by small biases and local variability in the lower atmospheric layers.

For these reasons, we believe that presenting the retrieval of the atmospheric profiles is an integral part of the study and can help colleagues when considering how to approach similar studies in future, so should not be overlooked.

We have added a discussion on this aspect and included the corresponding figure in section 3.1.3 of the manuscript.

**References**

Di Natale, G., Bianchini, G., Del Guasta, M., Ridolfi, M., Maestri, T., Cossich, W., Magurno, D., and Palchetti, L.: Characterization of the Far Infrared Properties and Radiative Forcing of Antarctic Ice and Water Clouds Exploiting the Spectrometer-LiDAR Synergy, Remote Sensing, 12, https://doi.org/10.3390/rs12213574, 2020.

Maestri, T., Rizzi, R., Tosi, E., Veglio, P., Palchetti, L., Bianchini, G., Girolamo, P. D., Masiello, G., Serio, C., and Summa, D.: Analysis of cirrus cloud spectral signatures in the far infrared, Journal of Geophysical Research, 141, 49–64, 2014.

Figure 1: Upper panel: the turquoise diamonds map the aircraft track over the Alomar site up to 08:25:00 UTC, 30 seconds before the first overpass. We use photographs taken at this time and at 08:27:00, 30 seconds after the overpass to estimate the drift of the aircraft contrail, shown by the green diamonds. Extrapolating this to 08:26:30 UTC indicates that at the time of the overpass the contrail released earlier in the flight, to the South-West, is directly over Alomar at the time of the overpass, this time also coincides with a distinct rise in radiance at 900 cm-1, observed by FINESSE. Middle and lower panels: reconstructed INCAS track with the corresponding photos of the contrail in the sky for the two sequences measure by FINESSE. In particular in the middle right panel is reported the increasing radiance at 900 cm-1 detected by FINESSE at the first sequence, when the contrail intersects the FINESSE's FOV.

Figure 2: Top panel: comparison of the effective diameters calculated from INCAS KA data (violet dots) withe the corresponding error and standard deviation between 08:19–09:00 UTC on 17 February 2023 and those retrieved from FINESSE (red triangles) between 08:25–08:38 UTC of the same day by assuming the habit distribution  $f_{1h}$  in Eq. 1. Second panel: same comparison but assuming  $f_{3h}$ . Third panel: same comparison but assuming  $f_{2h}$ . Bottom panel: same comparison but assuming  $f_{4h}$ . For brevity, the caption labels do not explicitly state the fraction of plates considered for crystal lengths below 50  $\mu$ m.

**Spectral Radiance Differences: (HBR+DX+PL) - (HBR+PL)**

Figure 3: Differences between radiances simulated by considering and non considering the droxtal contribution introduced in Eq. (1) in comparison with the FINESSE noise. Top left panel: considering OD = 0.05 and  $D_{ei}=20~\mu\mathrm{m}$ . Right top: considering OD = 0.2 and  $D_{ei}=20~\mu\mathrm{m}$ . Left bottom panel: OD = 0.05 and  $D_{ei}=30~\mu\mathrm{m}$ . Right bottom panel: considering OD = 0.2 and  $D_{ei}=30~\mu\mathrm{m}$ .

Figure 4: Upper panel:Information content of retrieval for the 4 habit distributions, corresponding to the different colors, considered in the analysis. Middle panel: degrees of freedom. Bottom pane: correlations  $OD-D_{ei}$ .

Figure 5: Upper panel:Information content of retrieval for the case FIR was considered (blue) and neglected (orange). Middle panel: degrees of freedom. Bottom pane: correlations  $OD-D_{ei}$ .

Figure 6: Top panel: comparison of the retrieved  $D_{ei}$  by including the FIR (blue dots) wit respect to the "truth" (red curve). Bottom panel: same but neglecting the FIR (green dots). In both cases the initial guess and a apriori is 80  $\mu$ m as done with the analysis.

**Scenario: Dei30\_OD0.18**

**Scenario: Dei30\_OD0.18\_650-1500cm-1**

Figure 7: Top panel: fit and residuals by using the FIR for the case OD = 0.18 and  $D_{ei}=30~\mu\mathrm{m}$  . Bottom panel: same by neglecting the FIR.

Figure 8: Left panel: vertical distribution of averaging kernels of temperature. Right panel: same for water vapour.

---

## Author Response (AR2)

**Answers to reviewers**

January 2, 2026

**REV1** The authors have done a nice job responding to the concerns the other reviewer and I had with the initial draft. There is one minor thing that needs adjusting. The middle panel of Figure 20, and the associated text, is (almost certainly) showing the correlation between the uncertainties in the retrieved Dei and OD – not the correlation between the actual retrieved values.

**ANSWER**

As we have added in the text in Section 5, the correlation between the cloud parameters OD and $D_{ei}$, as well as the IC and the DOFs, are computed at the end of the first retrieval iteration in order to quantify the intrinsic information content of the spectral measurements independently of retrieval convergence effects. Therefore, these are not correlations between the retrieved cloud parameters, but rather correlations between the parameters at the end of the first iteration of the inversion algorithm. We have also modified the caption of Figure 20 to clarify this point.

**REV2** I appreciate the authors' replies to my comments and some added analyses. Here are a few further comments: 1. The finding that temperature information is limited to 1km and water vapour 4km is interesting, and surprising - how can water vapour be constrained if temperature isn't? 2. Fig 20. Define DOF. I'm confused by what each panel shows and means here. Panel a: what state variables (and how many of them) is the IC assessed for? Panel c: how is DF quantified? Why does panel c show little difference while Panel a shows a larger difference? 3. Fig 21: can you clarify whether the "retrieved" states lead to a better closure in MIR than FIR - this isn't so clear concerning the water vapour lines (larger non-closure also noticed at the higher wavenumber end of the spectrum)! Is this due to the water vapour (uncertainty) or residual effect of habit fitting? 4. Regarding the cloud inhomogeneity - what does the in situ data say about this?

**ANSWER**

1. This difference in sensitivity is not surprising. Our previous work based on REFIR-PAD measurements at Dome-C, Antarctica — particularly Di Natale et al. (2017) — demonstrates, via singular value decomposition (SVD), that temperature sensitivity is mainly confined to the first kilometer above the surface, while water vapour sensitivity extends up to about 4 km. This is precisely the reason why the analysis of REFIR-PAD measurements is performed by limiting the number of fitted vertical levels accordingly. With the current retrieval setup, temperature is more constrained than water vapour, since it is retrieved using fewer fitted levels. Moreover, in both cases the a priori constraints are intentionally chosen to be sufficiently loose with respect to the actual atmospheric profiles, so as to prevent any undue over-constraining of the retrieval.

2. The degrees of freedom (DOFs) are computed as the trace of the averaging kernel matrix restricted to the two cloud parameters, namely the optical thickness (OD) and the effective diameter ($D_{ei}$), since the objective was precisely to assess the differences with respect to the cirrus analysis. The information content (IC) is computed as Shannon information content, i.e.

$$\text{IC} = \frac{1}{2} \ln \left| \mathbf{S}_a \mathbf{S}_x^{-1} \right| = \frac{1}{2} \ln \left| (\mathbf{I} - \mathbf{A})^{-1} \right| = \frac{1}{2} \sum_{i=1}^{2} \ln \left( \frac{1}{1 - \lambda_i} \right) \tag{1}$$

where $\mathbf{S}_a$ is the a priori covariance matrix and $\mathbf{S}_x$ is the posterior covariance matrix for the parameters $\mathbf{x} = (D_{ei}, OD)$ and the vertical bars denote the determinant, $\mathbf{A}$ is the averaging kernel matrix, $\mathbf{I}$ is the identity matrix, the vertical bars denote the determinant, and $\lambda_i$ are the eigenvalues of $\mathbf{A}$. The second equality follows from the relationship $\mathbf{S}_x = (\mathbf{I} - \mathbf{A})\mathbf{S}_a$, while the last equality is obtained by diagonalising the averaging kernel matrix and expressing the determinant as the product of its eigenvalues. IC and DOFs are evaluated at the first iteration of the retrieval, in order to quantify the intrinsic information content of the spectral measurements independently of retrieval convergence effects. As already mentioned, the degrees of freedom (DF) are computed as the trace of the averaging kernel matrix,

restricted to the two cloud parameters:

$$\text{DF} = \text{tr}(\mathbf{A}) = \sum_{i=1}^{2} \lambda_i \tag{2}$$

Although the difference in degrees of freedom between the two spectral configurations is small, the corresponding difference in information content is much larger. This is because the inclusion of far-infrared radiances significantly increases the sensitivity of the measurements to the retrieved parameters, reducing their dependence on the a priori. This effect is evident from the fact that the information content in Eq. 1 is computed as the logarithm of the inverse of (1 minus the eigenvalues of the averaging kernel), so as these eigenvalues approach 1, the information content increases rapidly. We have added the following sentence in Section 5 to clarify the procedure and the indicators used:

To explain this behaviour, we investigated three diagnostic indicators, namely the Shannon information content (IC), the number of degrees of freedom (DOFs), and the correlation between the cloud parameters, namely the optical thickness (OD) and the effective diameter ($D_{ei}$), since the objective was precisely to assess the differences with respect to the cirrus analysis. The IC is computed as (Rodgers, 2000):

$$\text{IC} = \frac{1}{2} \ln \left| \mathbf{S}_a \mathbf{S}_x^{-1} \right| = \frac{1}{2} \ln \left| (\mathbf{I} - \mathbf{A})^{-1} \right| = \frac{1}{2} \sum_{i=1}^{2} \ln \left( \frac{1}{1 - \lambda_i} \right) \tag{3}$$

where $\mathbf{S}_a$ is the a priori covariance matrix and $\mathbf{S}_x$ is the posterior covariance matrix for the parameters $\mathbf{x} = (D_{ei}, OD)$ and the vertical bars denote the determinant, $\mathbf{A}$ is the averaging kernel matrix, $\mathbf{I}$ is the identity matrix, the vertical bars denote the determinant, and $\lambda_i$ are the eigenvalues of $\mathbf{A}$. IC and DOFs are evaluated at the first iteration of the retrieval, i.e. around the a priori state, in order to quantify the intrinsic information content of the spectral measurements independently of retrieval convergence effects. The DOFs are computed as the trace of the averaging kernel matrix restricted to the two cloud parameters:

$$\text{DOF} = \text{tr}(\mathbf{A}) = \sum_{i=1}^{2} \lambda_i \tag{4}$$

3. The issue in the FIR portion, as already explained at line 483, is related to the difficulty in achieving an accurate characterisation of the instrument line shape in that spectral region. The residuals increase where $CO_2$ and $H_2O$ lines are very dense, as shown in Figure 21. They are not due to the presence of clouds; in fact, the degraded spectrum demonstrates that, in general, within the transparency micro-windows the residuals are smaller than or comparable to the noise level.

4. The INCAS KA probe data show that the variability in particle size occurs mainly along the vertical direction, as illustrated in Figure 11, which shows that the effective particle radius varies mostly between 20 and 50 $\mu$m with altitude. However, the radiative effect is determined by the integral contribution over the vertical distribution of particle sizes.

**References**

Di Natale, G., Palchetti, L., Bianchini, G., and Guasta, M. D.: Simultaneous retrieval of water vapour, temperature and cirrus clouds properties from measurements of far infrared spectral radiance over the Antarctic Plateau, Atmos. Meas. Tech., 10, 825–837, 2017.

Rodgers, C. D.: Inverse methods for atmospheric sounding : theory and practice, World Scientific Publishing, 2000.